# RAG without Forgetting:
# Continual Query-Infused Key Memory

Yuntong Hu [1][*]  Sha Li [2][*]  Naren Ramakrishnan [2]  Liang Zhao [1]

## Abstract

Retrieval-augmented generation (RAG) systems commonly improve robustness via query-time adaptations such as query expansion and iterative retrieval. While effective, these approaches are inherently stateless: adaptations are recomputed for each query and discarded thereafter, precluding cumulative learning and repeatedly incurring inference-time cost. Index-side approaches like key expansion introduce persistence but rely on offline preprocessing or heuristic updates that are weakly aligned with downstream task utility, leading to semantic drift and noise accumulation. We propose *Evolving Retrieval Memory (ERM)*, a training-free framework that transforms transient query-time gains into persistent retrieval improvements. ERM updates the retrieval index through correctness-gated feedback, selectively attributes atomic expansion signals to the document keys they benefit, and progressively evolves keys via stable, norm-bounded updates. We show that query and key expansion are theoretically equivalent under standard similarity functions and prove convergence of ERM's selective updates, amortizing optimal query expansion into a stable index with zero inference-time overhead. Experiments on BEIR and BRIGHT across 13 domains demonstrate consistent gains in retrieval and generation, particularly on reasoning-intensive tasks, at native retrieval speed.

## 1. Introduction

Retrieval-Augmented Generation (RAG) has emerged as a leading paradigm for knowledge-intensive tasks by grounding generation in retrieved documents (Lewis et al., 2020;

---

[*]Equal Contribution. [1]Emory University [2]Virginia Tech. Correspondence to: Yuntong Hu <yuntong.hu@emory.edu>, Liang Zhao <liang.zhao@emory.edu>.

*Proceedings of the 43$^{rd}$ International Conference on Machine Learning*, Seoul, South Korea. PMLR 306, 2026. Copyright 2026 by the author(s).

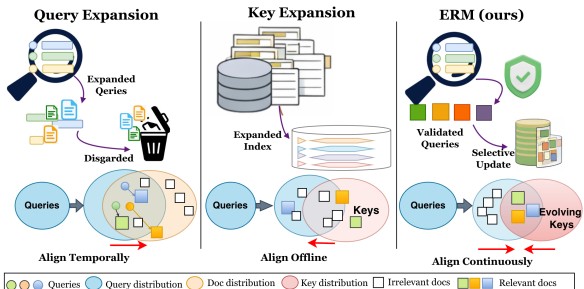

*Figure 1.* **Comparison of Query Expansion (QE), Key Expansion (KE), and Evolving Retrieval Memory (ERM).** *Left:* QE aligns queries to document space via inference-time expansions that are discarded after each query. *Middle*: KE persistently aligns documents to queries through offline index enrichment but incurs high cost and drift. *Right:* ERM converts validated query expansions into stable key updates, progressively aligns query and document distributions with no inference-time overhead.

Guu et al., 2020). Despite enhancing factual accuracy and domain adaptability, end-to-end RAG performance remains constrained by retrieval quality, which upper-bounds downstream generation. In practice, **query–corpus misalignment**, caused by semantic gaps between queries and indexed documents, ambiguous or underspecified query intent, and embedding anisotropy (Zhou et al., 2024) often leads to incomplete or imprecise retrieval, resulting in a dominant recall bottleneck.

To mitigate, prior work has explored query expansion (QE) (Zhang et al., 2024; Ma et al., 2023b; Koo et al., 2024) by enriching query representations with paraphrases or auxiliary semantic signals to increase overlap with relevant documents. While empirically effective, QE adapts at inference time: expansions are recomputed for each query and discarded afterward, yielding only transient benefits. Recent advances of adaptive (Jeong et al., 2024), iterative (Jiang et al., 2025) and agentic RAG (Li et al., 2025c) interleave retrieval with reasoning to enable multi-step search, reflection, and correction (Jeong et al., 2024; Jiang et al., 2025; Li et al., 2025c). However, despite improved per-query robustness, these methods remain fundamentally stateless, as retrieval adaptations do not persist across queries, leading to repeated computation and limited cumulative learning.

As a complementary axis, key expansion (KE) (Yang et al., 2025; Chen et al., 2024b) enriches document representations

in the retrieval index, but it is applied offline and query-agnostic, ignoring the online query distribution and downstream task utility, thus is prone to semantic drift and noise accumulation. Besides, the computational and storage overhead of precomputing enriched keys is significant, particularly for large-scale corpora, restricting scalability. Memory-augmented RAG addresses these limitations by introducing structured or associative memory modules (Jimenez Gutierrez et al., 2024; Gutiérrez et al., 2025) to store retrieved knowledge or intermediate reasoning states. This paradigm improves over traditional KE- and QE-RAG in terms of incorporating structured context. While promising, existing memory-augmented RAG systems often rely on heuristic or offline memory updates (Pan et al., 2025; Qian et al., 2025; Zhou et al., 2025) that are weakly aligned with downstream task utility, treat retrieval and memory maintenance as disjoint processes (Zhou et al., 2025), and lack principled mechanisms for memory selection, consolidation, and forgetting (Fu et al., 2025). These limitations hinder scalability, long-term robustness, and effective cumulative learning.

The above limitations of prior approaches stem not from isolated design choices, but from a structural separation between where adaptation occurs and how it is validated. Query-centric (query-expansion) methods adapt in response to the current query and generation context, yet discard these adaptations once the query terminates, leading to repeated computation and no cumulative learning. Index-centric (key-expansion) and memory-augmented methods, meanwhile, introduce persistence but operate largely offline or with weak supervision, without direct alignment to task-level success. This disconnect prevents principled credit assignment: *systems lack a reliable signal for deciding which retrieval behaviors are worth remembering and which should be forgotten.*

Consequently, effective long-term retrieval improvement requires jointly reasoning over query-time signals and index-time persistence. Retrieval memory must evolve selectively, guided by verifiable correctness signals, and incrementally, without destabilizing the retrieval space or incurring inference-time overhead. This observation motivates our proposal for *Evolving Retrieval Memory (ERM)* (Figure 1), which explicitly bridges transient query expansions and persistent key representations through correctness-gated updates, selective attribution, and stable accumulation. By tightly coupling online adaptation with validated memory evolution, ERM transforms per-query gains into amortized improvements, overcoming the statelessness of query-centric methods and the drift-prone nature of offline memory construction.

Our contributions can be summarized as follows:

- We propose Evolving Retrieval Memory (ERM), a training-free RAG framework that converts transient query-time adaptations into persistent retrieval improvements by correctness-gated feedback, selectively attributing atomic expansion signals to the keys they benefit, and progressively evolving the retrieval index through stable, norm-bounded updates without model retraining.

- We theoretically prove that query and key expansions are equivalent under standard similarity functions, and show that our bounded, selective key updates provably converge, amortizing the benefits of optimal query expansion into a stable index with zero inference-time cost.

- We evaluate ERM on BEIR and BRIGHT across 13 domains, with multiple retrievers, and indexing strategies for retrieval and RAG. ERM consistently improves retrieval and generation performance, with pronounced gains on reasoning-intensive tasks. By consolidating successful retrieval patterns and enabling query-conditional key specialization, ERM scales predictably with experience while operating at native retrieval speed, substantially outperforming LLM-based query expansion in efficiency.

## 2. Related Work

### 2.1. Memory-Augmented Retrieval

Memory-augmented retrieval introduces explicit memory architectures (Packer et al., 2023; Zhong et al., 2024; Chhikara et al., 2025; Fang et al., 2025; Xu et al., 2025) that persist, organize, and adaptively reuse information from prior interactions or reasoning steps, improving retrieval quality and efficiency (Du et al., 2025; Trivedi et al., 2023; Wang et al., 2025). Unlike static corpus retrieval, these methods maintain evolving memory states that support long-horizon reasoning. Existing approaches differ primarily in memory source and granularity. Segment-level or working memory stores compressed intermediate contexts or evidence (Pan et al., 2025; Yan et al., 2025), query-conditioned memory selectively activates relevant past knowledge via cues or signals (Fu et al., 2025; Liao, 2025), and global or parametric memory constructs long-term representations distilled from accumulated experience (Qian et al., 2025). To capture structural dependencies, several works adopt graph-structured memory, enabling relational retrieval and multi-hop reasoning over entities and evidence (Liu et al., 2024a; Li et al., 2025a; Jimenez Gutierrez et al., 2024; Gutiérrez et al., 2025; Zhou et al., 2025).

### 2.2. Retrieval-Augmented Generation (RAG)

RAG enhances LLM performance for knowledge-intensive tasks (Izacard et al., 2023; Lewis et al., 2020; Gao et al., 2023b; Tang & Yang, 2024; Xiong et al., 2024; Fan et al., 2024) by incorporating external knowledge. Early effort integrates retrieval into next-token prediction (Khandelwal et al., 2020; Liu et al., 2024b; Izacard & Grave, 2020; Wang

et al., 2023), or end-to-end pipeline (Borgeaud et al., 2022; Izacard et al., 2023; Guu et al., 2020) to jointly optimize the retriever and generator. Research incorporates modules like query refinement (Chan et al., 2024; Gao et al., 2023a; Ma et al., 2023a), post-retrieval processing (Abdallah et al., 2025; Xu et al., 2023) to improve the retrieval quality, and enhance the efficient use of retrieved contents (Jiang et al., 2023; Mao et al., 2023; 2024; Qian et al., 2024; Yan et al., 2024). Structure-augmented RAG involves tree (Fatehkia et al., 2024; Li et al., 2025d; Hu et al., 2025), graph (Sarthi et al., 2024; Edge et al., 2024; Guo et al., 2024) and hypergraph (Feng et al., 2025; Luo et al., 2025; Wang & Han, 2025) to capture relational and multi-hop context and enhance retrieval relevance. RAG synergize with reasoning capabilities for complex tasks (Jeong et al., 2024; Lee et al., 2024; Asai et al., 2024; Shao et al., 2023). Recent advance in DeepResearcher (Zheng et al., 2025; Li et al., 2025b; Huang et al., 2025) extends RAG by embedding retrieval into a broader, agentic research workflow that combines LLM reasoning with dynamic tool use to solve open-ended, complex tasks.

## 2.3. Query Expansion and Rewriting

Query expansion and rewriting methods improve retrieval by transforming the original query into a more effective search representation. Classical approaches use pseudo-relevance feedback (Amati & Van Rijsbergen, 2002) or lexical expansion via synonyms and related terms. Neural methods rewrite queries using sequence-to-sequence models (Ma et al., 2023a) or generate hypothetical documents that serve as enriched query representations, as in HyDE (Gao et al., 2023a). Diversified expansion strategies such as Diver produce multiple reformulations to cover different facets of the information need. BGE-Reasoner-Rewriter (Lan et al., 2025) combines retrieval-oriented reasoning with query rewriting in a single model. While these methods improve retrieval at inference time, their gains are transient: each query is expanded independently without accumulating knowledge from prior successful retrievals. ERM differs fundamentally by converting these transient query-time signals into persistent key-side improvements, enabling the retrieval index itself to evolve.

## 3. Problem Formulation

**Retrieval Systems.** Let $D = \{d_1, \ldots, d_n\}$ be a corpus of $n$ documents and let $q \in \mathcal{Q}$ denote a user query. Each document $d_i$ is indexed by a key $k_i \in \mathcal{K}$, yielding the key set $K = \{k_i\}_{i=1}^{n}$, where $\mathcal{K}$ denotes the retriever's representation space. A retrieval system is defined by a similarity function such as an inner-product:

$$S : \mathcal{Q} \times \mathcal{K} \to \mathbb{R}, \tag{1}$$

which scores every key against the query. We denote by $R_K(q) \subset [n]$ the index set of the $N$ documents whose keys achieve the highest similarity to $q$.

In practice, document keys are constructed using various key-building strategies, such as titles, keywords, abstracts, full document content, or document chunking, with the goal of filtering low-information terms and improving retrieval effectiveness. Key construction is typically performed globally during an offline corpus indexing stage and remains independent of individual online queries.

**Retrieval-Augmented Generation (RAG).** A RAG system augments an LLM-based generator $\pi$ with a retriever. Given a query $q$, the retriever returns the top-$N$ relevant documents $R_K(q)$, whose values are passed as additional context to the generator, yielding

$$Y_K(q) \sim \pi\big(\cdot \mid q, \{d_i\}_{i \in R_K(q)}\big), \tag{2}$$

where $Y_K(q) \in \mathcal{Y}$ denotes the generated response conditioned on the query and the document keys $K$.

**Query Expansion.** Query Expansion (QE) augments the original query with additional related terms or reformulations to improve retrieval coverage by a query expansion method $\phi$. Formally, given a query $q \in \mathcal{Q}$, a query expansion method $\phi$ produces a set of expansion units

$$c(q) = \{e_1, \ldots, e_m\} \sim \phi(\cdot \mid q), \tag{3}$$

yielding an expanded query representation $q' := \{q\} \cup c(q)$. Retrieval is then performed using the expanded query. In modern retrieval and RAG systems, the expanded query is discarded after inference and induces no persistent modification to the retrieval memory.

Although both query expansion and corpus key building aim to improve retrieval performance, they exhibit fundamental limitations:

- Query expansion provides only transient gains and requires repeated augmentation at inference time;
- Key building is performed offline and globally, remaining fixed thereafter and ignoring the actual online query distribution;
- Corpus keys, query distributions, and downstream task objectives are decoupled, preventing retrieval system from being guided by task-level feedback.

These limitations motivate a retrieval mechanism that is online, query-driven, efficient, and selectively updated only when validated by downstream task performance.

## 4. Evolving Retrieval Memory

Modern retrieval systems can refresh corpus keys via periodic re-embedding or re-indexing, but such updates are

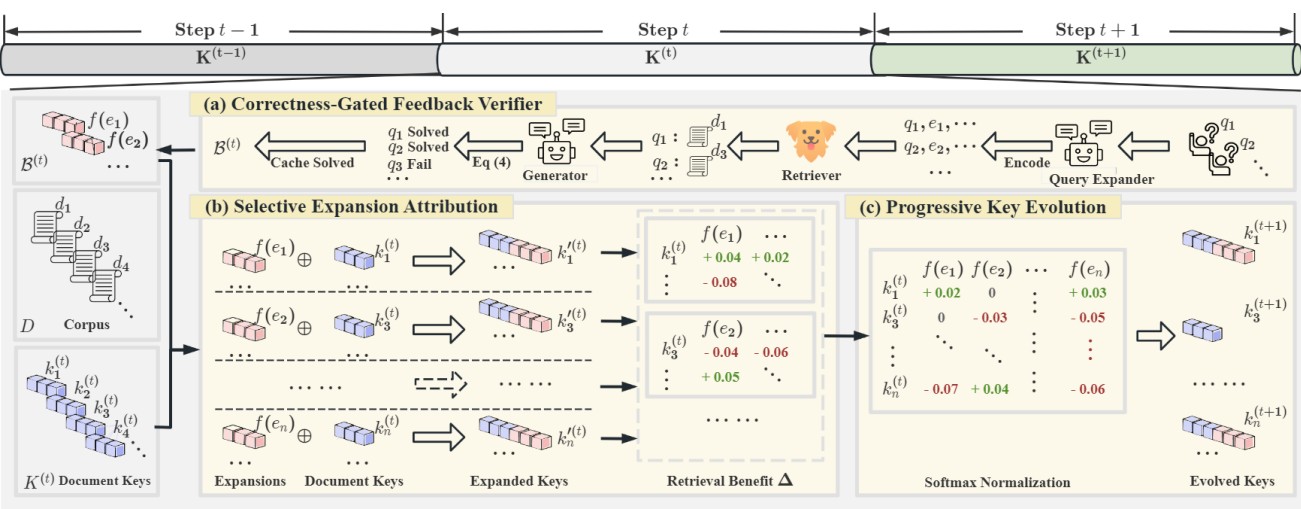

Figure 2. **Illustration of ERM**. (a) Correctness-gated verification filters task-validated query expansion units. (b) Selective attribution assigns expansion benefits to retrieved documents. (c) Softmax-normalized accumulation updates document keys.

typically corpus-wide and incur cost that scales with the corpus size. Meanwhile, query expansion (QE) improves retrieval using query-specific signals, but the gains are transient because expansions are discarded after inference. A key observation is that real-world queries are highly long-tailed and Zipf-like (Beitzel et al., 2004; Downey et al., 2008; Zhang & Ross, 2020), with repeated queries/intents concentrating traffic on a small fraction of the corpus. This property implies that *frequent global key updates are unnecessary* and that *applying query expansion uniformly at inference time is inefficient*. Together, these observations motivate a lazy update strategy: keys should be updated only where repeated, task-solved queries provide reliable evidence. Such selective updates yield amortized query-scale cost that is independent of the corpus size in expectation. Accordingly, we propose *Evolving Retrieval Memory (ERM)*, a training-free retrieval enhancement framework that *selectively evolves* document keys using task-validated query expansion signals. Specifically,

- Section 4.1 introduces the *Correctness-Gated Feedback Verifier*, which leverages feedback from RAG systems to decide when memory evolution should be triggered (Figure 2 (a)). This mechanism generalizes across retrieval methods and RAG scenarios.
- Section 4.2 presents *Selective Expansion Attribution* to decompose query expansions into atomic semantic attributions and assigns each attribution only to keys whose relevance is improved, bridging query expansion and key building (Figure 2 (b)).
- Section 4.3 introduces *Progressive Key Evolution* as shown in (Figure 2 (c)), an efficient accumulation and update mechanism that enables keys to evolve in a stable, batched manner without retraining any model parameters.

### 4.1. Correctness-Gated Feedback Verifier

A central question in ERM is: *when should query expansions be cached and used to update keys?* retrieval related tasks such as RAG are evaluated either by downstream task metrics, or both retrieval and downstream task metrics. In practice, different datasets present different supervision regimes: (1) full retrieval and downstream task ground truth, (2) downstream task-only supervision, or (3) no explicit ground truth (human-as-judge or LLM-as-judge regimes).

**Unified correctness signal.** Let $V_r$ denote a retrieval verifier (e.g., recall@K or DPR match), and let $V_g$ denote a generation verifier (e.g., ROUGE, task loss, or LLM-as-judge scoring). Both $V_r$ and $V_g$ output binary correctness indicators. Each underlying metric is mapped to $\{0, 1\}$ by a task-specific threshold or decision rule.

Given an expanded query representation $q' := \{q\} \cup c(q)$, where $c(q) = \{e_1, \ldots, e_m\}$ denotes the set of expansion units produced by a query expansion method, we define a unified success indicator:

$$\text{success}(q', K) := \mathbf{1}\big[V_r\big(R_K(q')\big) + V_g\big(Y_K(q')\big) \geq 1\big].$$
(4)

The expanded query is accepted if either retrieval correctness or generation correctness holds, preventing ERM from overfitting to a single subsystem. Only expansion units validated by this correctness signal are cached for memory evolution, thereby avoiding harmful or noisy updates. We next introduce how these validated units are attributed to individual document keys.

## 4.2. Selective Expansion Attribution

Even when an expansion is globally useful, not all expansion components should update all retrieved documents. ERM addresses this through a similarity–gain attribution mechanism that routes each expansion unit only to the keys for which it provides positive retrieval benefit.

For each retrieved document $d_i$ by $R_K(q')$ and each expansion unit $e_j \in c(q)$, we measure the marginal benefit of assigning $e_j$ to the document key with respect to the original query $q$:

$$\Delta_{i,j}(q) \,=\, \mathrm{sim}(f(q), k_i \oplus f(e_j)) - \mathrm{sim}(f(q), k_i)\,, \quad (5)$$

where $\oplus$ denotes a retriever-compatible key augmentation operator (additive composition: $k_i \oplus v = k_i + v$) that appends the expansion representation $f(e_j)$ to the key of document $d_i$, $f(\cdot)$ maps queries and expansion units into the retriever's representation space, and $\mathrm{sim}(\cdot, \cdot)$ is the retrieval similarity function. This formulation is grounded in the following equivalence between query-side and key-side expansion:

**Proposition 4.1** (Query-Key Equivalence Under Semantic Composition). *Assume the retriever similarity $\mathrm{sim}(\cdot, \cdot)$ is bilinear or monotone under additive embeddings. Then for any expansion unit $e_j$,*

$$\mathrm{sim}(f(q + e_j), k_i) \,\propto\, \mathrm{sim}(f(q), k_i \oplus f(e_j)).$$

*Thus expanding queries is equivalent to expanding keys with respect to retrieval ranking under standard similarity operators.*

While correctness is verified using the expanded query $q'$, attribution is computed relative to $q$ to capture the marginal contribution of each expansion unit. We retain an expansion unit for a given key only if it yields a positive retrieval gain over the original query, i.e., $\Delta > 0$.

ERM operates through two complementary mechanisms: (1) *Hot cache reinforcement*: For frequently queried documents, ERM strengthens the association between document keys and successful query patterns, ensuring that similar future queries reliably retrieve relevant documents regardless of query expansion variability; (2) *Relative ranking improvement*: For long-tail documents that receive fewer key updates, their relative similarity to queries increases as other documents' keys become more specialized, indirectly improving retrieval coverage.

## 4.3. Progressive Key Evolution

To reduce drift from noisy or query-specific expansions and to prevent the accumulation of irrelevant correlations, ERM adopts a strategy of batched, progressive evolution. For each document $d_i$, ERM maintains an expansion memory $u_i$ that stores expansion units accumulated from past task-solved queries. Given a batch of queries $\mathcal{B}$, ERM computes query-local attribution weights for each query $q \in \mathcal{B}$ to normalize the relative contribution of its expansion units:

$$w_{i,j}(q) = \mathrm{softmax}_{e_j \in c(q)}(\Delta_{i,j}(q)), \quad (6)$$

where the softmax is taken over the expansion units generated from the same query. These normalized contributions are then aggregated across the batch to update the persistent relevance scores:

$$s_{i,j} \leftarrow s_{i,j} + \sum_{q \in \mathcal{B}} w_{i,j}(q)\,\Delta_{i,j}(q), \quad (7)$$

where $s_{i,j}$ denotes the accumulated relevance score of expansion unit $e_j$ for document $d_i$. Each updated pair $(e_j, s_{i,j})$ is inserted into the expansion memory $u_i$, discarding lower-scoring entries when the capacity constraint is exceeded.

We show that this scoring mechanism identifies the most beneficial expansion units:

**Proposition 4.2** (Cumulative Consistency of Attribution Scores). *Fix a document $d_i$ and suppose its key $k_i$ remains unchanged while ERM processes a sequence of queries $\{q_t\}_{t=1}^{T}$ drawn i.i.d. from $\mathcal{Q}$. For each expansion unit $e_j$, define the expected gated attribution gain $\mu_{i,j} := \mathbb{E}_{q \sim \mathcal{Q}}[\mathbf{1}_{i,j}(q)\,w_{i,j}(q)\,\Delta_{i,j}(q)]$. Then $s_{i,j}^{(T)}/T \xrightarrow{\mathrm{a.s.}} \mu_{i,j}$ as $T \to \infty$. Moreover, if the $x$-th and $(x+1)$-th largest values among $\{\mu_{i,j}\}_j$ are separated by a nonzero gap, the top-$x$ selection operator $\tau_x(u_i)$ converges almost surely to the set of $x$ expansion units with the largest $\mu_{i,j}$.*

*Proof.* See Appendix A.1.

When evolving document keys, ERM augments each key using the highest-scoring expansion units retained in $u_i$:

$$k_i^{(t+1)} \,=\, k_i^{(t)} \oplus f\left(\tau_x(u_i)\right), \quad (8)$$

where $\tau_x(u_i)$ returns $x$ expansion units with the largest accumulated relevance scores. This design ensures that key evolution incorporates only strongly supported semantic evidence, while the expansion memory prevents uncontrolled growth. We establish that the resulting key sequences converge to a stable index:

**Theorem 4.3** (Stability of Key Sequences). *Suppose expansion embeddings are norm-bounded, $\|f(e_j)\| \leq M$ for all $j$. (i) In the offline setting, only finitely many evolution rounds are executed and each key converges to $k_i^* = k_i^{(0)} \oplus f(\tau_x^*)$. (ii) In the online setting, if the per-round key displacement satisfies $\sum_{r=0}^{\infty} \|k_i^{(r+1)} - k_i^{(r)}\| < \infty$, then $\{k_i^{(r)}\}$ converges to a limit $k_i^*$; ERM's saturation criterion ensures this condition holds by terminating each round once the marginal benefit diminishes.*

*Table 1.* **Retrieval performance with and without ERM across BEIR and BRIGHT benchmarks.** We report nDCG@1 for all datasets, with Δ rows showing percentage improvement from ERM. Results span sparse (BM25), open-source dense (BGE, GTE, MiniLM), and proprietary (Cohere, Voyage) retrievers. ERM consistently improves performance, with the largest gains on reasoning-intensive datasets (LeetCode, Pony, AoPS, TheoremQA).

| Model | BEIR | | StackExchange | | | | | | | Coding | | Theorem-based | | Avg. |
|---|---|---|---|---|---|---|---|---|---|---|---|---|---|---|
| | NFC. | Sci. | Bio. | Earth. | Econ. | Psy. | Rob. | Stack. | Sus. | Leet. | Pony | AoPS | TheoT. | |
| *Sparse model* | | | | | | | | | | | | | | |
| BM25 | 20.7 | 35.4 | 46.6 | 42.2 | 27.2 | 31.7 | 22.8 | 27.4 | 28.7 | 13.4 | 36.6 | 0.9 | 7.9 | 26.3 |
| BM25 (w/ ERM) | 25.5 | 38.4 | 68.6 | 47.7 | 38.8 | 49.3 | 24.1 | 37.9 | 35.6 | 16.0 | 60.0 | 20.7 | 37.8 | 38.5 |
| Δ | +23% | +8% | +47% | +13% | +43% | +56% | +6% | +38% | +24% | +19% | +64% | +2200% | +378% | +46% |
| *Dense Model* | | | | | | | | | | | | | | |
| BGE-Large | 25.5 | 44.7 | 95.1 | 78.8 | 58.2 | 53.5 | 43.4 | 50.5 | 79.1 | 17.9 | 43.1 | 8.7 | 33.5 | 48.6 |
| BGE-Base | 28.2 | 42.8 | 96.0 | 79.6 | 62.1 | 51.5 | 45.4 | 51.3 | 79.1 | 15.8 | 57.8 | 7.0 | 24.5 | 49.3 |
| BGE-M3-Dense | 21.3 | 37.2 | 81.5 | 73.5 | 48.4 | 41.4 | 46.3 | 43.7 | 67.8 | 15.8 | 68.3 | 6.1 | 28.3 | 44.6 |
| GTE-Base | 24.5 | 45.6 | 97.8 | 77.9 | 65.3 | 54.8 | 44.8 | 57.2 | 76.3 | 17.2 | 57.2 | 2.5 | 33.8 | 49.9 |
| MiniLM | 24.6 | 44.6 | 93.1 | 79.5 | 49.3 | 46.4 | 41.3 | 51.3 | 72.5 | 18.5 | 52.7 | 7.0 | 33.6 | 47.3 |
| BGE-Large (w/ ERM) | 26.5 | 45.1 | 91.3 | 80.4 | 58.8 | 55.1 | 40.4 | 55.6 | 75.9 | 25.8 | 87.5 | 20.1 | 61.3 | 55.7 |
| Δ | +4% | +1% | -4% | +2% | +1% | +3% | -7% | +10% | -4% | +44% | +103% | +131% | +83% | +15% |
| GTE-Base (w/ ERM) | 25.0 | 47.9 | 94.9 | 82.6 | 64.0 | 65.2 | 43.9 | 58.9 | 73.2 | 24.1 | 73.2 | 15.2 | 65.6 | 56.4 |
| Δ | +2% | +5% | -3% | +6% | -2% | +19% | -2% | +3% | -4% | +40% | +28% | +508% | +94% | +13% |
| *Proprietary models* | | | | | | | | | | | | | | |
| Cohere | 33.3 | 45.5 | 97.1 | 79.6 | 56.6 | 63.0 | 43.2 | 56.4 | 66.7 | 14.8 | 38.4 | 8.1 | 28.9 | 48.7 |
| Voyage | 35.1 | 48.0 | 98.2 | 84.0 | 64.0 | 66.3 | 42.9 | 60.2 | 72.9 | 13.1 | 41.0 | 8.6 | 30.7 | 50.8 |
| Cohere (w/ ERM) | 34.2 | 47.1 | 95.8 | 81.2 | 58.4 | 65.7 | 40.8 | 59.1 | 68.3 | 18.2 | 76.5 | 17.8 | 54.2 | 55.2 |
| Δ | +3% | +4% | -1% | +2% | +3% | +4% | -6% | +5% | +2% | +23% | +99% | +120% | +88% | +13% |
| Voyage (w/ ERM) | 36.0 | 49.8 | 97.1 | 85.3 | 65.8 | 68.9 | 41.5 | 63.0 | 74.1 | 16.8 | 81.2 | 18.5 | 53.6 | 56.3 |
| Δ | +3% | +4% | -1% | +2% | +3% | +4% | -3% | +5% | +2% | +28% | +98% | +115% | +75% | +11% |

*Proof.* See Appendix A.2.

**Corollary 4.4** (Expected Consistency Under Additive Similarity)**.** *Under inner-product similarity* $\text{sim}(a, b) = a^\top b$ *and additive key augmentation* $k_i \oplus v = k_i + v$, *the converged top-$x$ selection $S^*$ satisfies $S^* = \arg\max_{|S|=x} \sum_{j \in S} \mu_{i,j}$, i.e., it maximises expected similarity gain for future queries.*

*Proof.* See Appendix A.3.

The branch of expanded queries may arise from offline pre-processing or online streaming. In the online setting, after each batch update the buffer $\mathcal{B}$ is cleared and processing continues. To determine whether the current branch remains informative, ERM evaluates its alignment gain:

$$\Delta^{\mathcal{B}} = \max_{(q, c(q')) \in \mathcal{B}} \max_{i,j} \Delta_{i,j}(q). \qquad (9)$$

ERM maintains the best branch gain observed so far, denoted by $\max\{\Delta^{\mathcal{B}_1}, \ldots, \Delta^{\mathcal{B}_t}\}$. Key evolution terminates via a patience-based saturation criterion: if $P$ consecutive branch extensions fail to exceed a relative margin $\rho$ of the best historical gain,

$$\Delta^{\mathcal{B}_{t+1}} \leq (1 - \rho) \max\{\Delta^{\mathcal{B}_1}, \ldots, \Delta^{\mathcal{B}_t}\}, \qquad (10)$$

the index is deemed saturated and a batch evolution step is triggered, preventing premature stopping from transient gain fluctuations. After the key evolution phase completes, generation proceeds as

$$Y_{K'}(q) = G\big(q, \{v_i\}_{i \in R_{K'}(q)}\big), \qquad (11)$$

where the query $q$ is executed directly against the evolved document keys $K'$ at inference time. The updated keys provide high-recall retrieval without requiring further query expansion, yielding both improved generation quality and reduced inference latency.

Finally, we show that ERM's expansion cost is sublinear in the query volume under realistic query distributions:

**Proposition 4.5** (Amortized Expansion Cost)**.** *Assume queries are drawn i.i.d. from a Zipf-like intent distribution with exponent $\alpha > 1$. Then ERM's cumulative expansion cost grows as $\Theta(T^{1/\alpha})$, sublinearly in the query volume $T$, and its per-query amortized cost vanishes as $T \to \infty$.*

*Proof.* See Appendix A.4.

## 5. Empirical Evaluation

**Dataset.** We evaluate our method on 13 datasets from the BRIGHT (Su et al., 2024) and BEIR (Thakur et al., 2021) benchmarks. Detailed dataset descriptions are provided in Appendix B.1.

*Table 2.* **End-to-end question-answering performance with ERM-augmented retrieval.** We evaluate answer quality using Claude-3.5-sonnet for both generation and evaluation on StackExchange Q&A domains. ERM improves generation quality across all retrievers, with BM25 showing the largest gain (+6% average) and dense retrievers achieving +2–4% improvements, demonstrating that improved retrieval translates to better downstream task performance.

| Retriever | Bio. | Earth. | Econ. | Psy. | Rob. | Stack. | Sus. | Average |
|---|---|---|---|---|---|---|---|---|
| BM25 | 71.8 | 74.5 | 69.8 | 74.8 | 71.4 | 76.4 | 69.5 | 72.6 |
| BM25 (w/ ERM) | 79.9 | 77.8 | 73.0 | 76.1 | 76.4 | 80.5 | 72.5 | 76.6 |
| Δ | +11% | +4% | +5% | +2% | +7% | +5% | +4% | +6% |
| BGE-Large | 80.2 | 79.3 | 67.6 | 69.1 | 74.8 | 78.5 | 72.2 | 74.5 |
| GTE-Base | 80.5 | 81.6 | 74.4 | 77.5 | 75.3 | 79.7 | 73.1 | 77.4 |
| Cohere | 81.1 | 82.9 | 76.6 | 78.7 | 79.9 | 82.1 | 73.7 | 79.3 |
| BGE-Large (w/ ERM) | 81.2 | 80.6 | 73.7 | 76.9 | 76.4 | 81.3 | 73.1 | 77.6 |
| Δ | +1% | +2% | +9% | +11% | +2% | +4% | +1% | +4% |
| GTE-Base (w/ ERM) | 82.2 | 80.9 | 75.1 | 79.9 | 76.9 | 84.6 | 73.6 | 79.0 |
| Δ | +2% | -1% | +1% | +3% | +2% | +6% | +1% | +2% |
| Cohere (w/ ERM) | 81.9 | 82.3 | 77.8 | 81.2 | 78.7 | 86.9 | 74.9 | 80.5 |
| Δ | +1% | -1% | +2% | +3% | -2% | +6% | +2% | +2% |

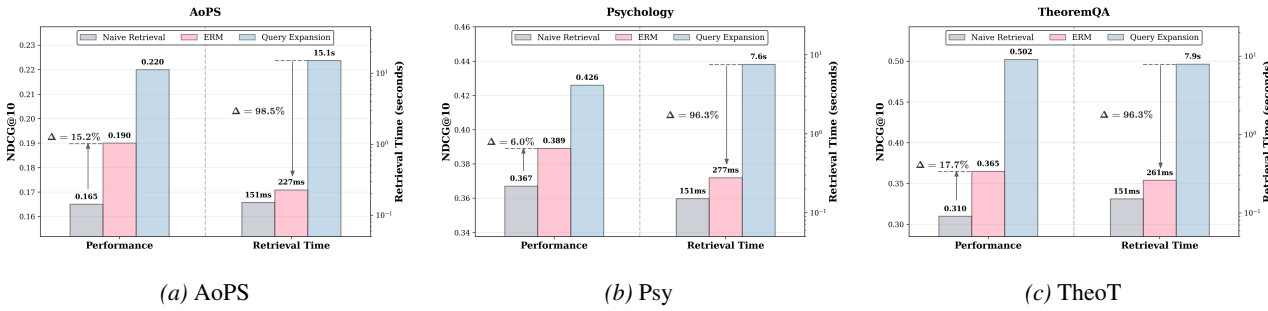

*(a)* AoPS       *(b)* Psy       *(c)* TheoT

*Figure 3.* **Performance vs. latency trade-off.** Comparison of retrieval performance (nDCG@10, left bars) and inference time (log scale, right bars) for Naive Retrieval, ERM, and Query Expansion (HyDE). ERM achieves performance competitive with or exceeding HyDE while maintaining near-native retrieval latency (ms vs. seconds). Configuration: GTE-base retriever, 0.5 split rate, title indexing.

**Evaluation Metrics.** We report normalized Discounted Cumulative Gain at rank 10 (nDCG@10) as the primary retrieval metric and Mean Reciprocal Rank (MRR) as a secondary metric, following BEIR and BRIGHT conventions. For generation tasks, we use the official evaluation metrics provided by the BRIGHT benchmark.

**Baselines.** We include representative query rewriting and expansion approaches, including BGE-Reasoner-Rewriter (Lan et al., 2025), clarification and specification, T5L-Turbo Query Rewriting (Ma et al., 2023b), contrastive-, facet-, and synonym-based expansion, HyDE query transformation (Gao et al., 2023a), and query decomposition. Experiments are conducted in a standard RAG pipeline with multiple index representations (title, keywords, abstract, and full text) and a diverse set of retrievers, including sparse methods such as BM25 (Robertson et al., 1995) and BGE-M3 (Chen et al., 2024a), dense retrievers such as BGE-Large/Base (Xiao et al., 2024), GTE-Base (Li et al., 2023), and MiniLM, as well as proprietary embedding APIs (Cohere and Voyage). Details are provided in Appendix B.1.

## 5.1. Main Results

We evaluate ERM under a repeated holdout adaptation protocol, varying the fraction of queries used for key evolution from 0.3 to 0.8 (in increments of 0.1). For each split, document keys are reset to their initial state, a randomly sampled subset of queries is used to perform adaptive key updates, and retrieval performance is measured on the remaining queries without further modification of the index. We combine multiple query expansion strategies and random seeds to assess robustness across different query orderings and adaptation subsets. Reported results aggregate performance across all tested configurations. Comprehensive per-dataset results and ablation studies are provided in Appendix B.

**Consistent gains across domains and retrievers.** As shown in Table 1, ERM consistently improves retrieval across all domains and retriever architectures, achieving +46% average nDCG improvement for BM25 and +13–15% for dense retrievers. Gains are particularly pronounced on reasoning-intensive datasets where surface-level lexical overlap is low: LeetCode (+19–44%), Pony (+64–103%), AoPS (+115–2200%), and TheoremQA (+75–378%). The

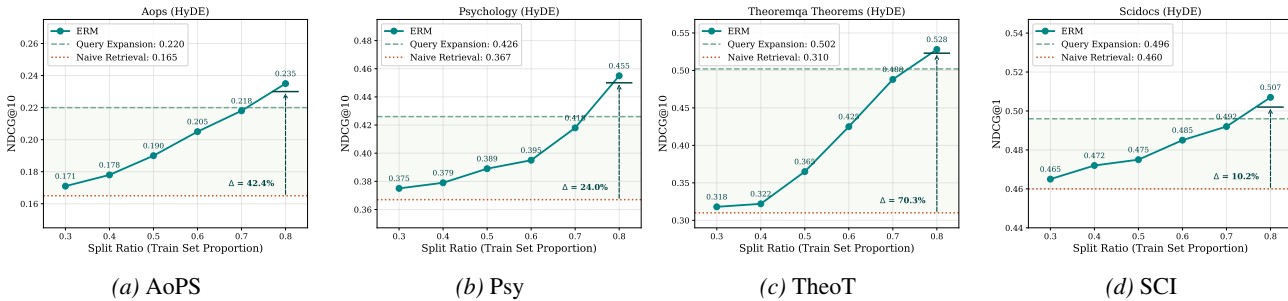

*Figure 4.* **ERM performance as a function of adaptation budget.** nDCG@10 on held-out queries after evolving keys using an increasing fraction (0.3–0.8) of disjoint adaptation queries, with keys reset for each split. Results are shown for the GTE-base retriever with HyDE query expansion and title-based indexing. Performance improves monotonically as ERM is allowed to adapt using more past queries.

largest relative gains occur for weaker retrievers (BM25: +46%), while already-strong proprietary models (Voyage) still gain +11%, indicating that ERM complements rather than duplicates the underlying retrieval capability.

**Downstream generation quality.** As shown in Table 2, ERM improves question-answering performance across all retrievers, with BM25 showing +6% average gain and dense retrievers achieving +2–4%. Even when retrieval metrics show modest gains (e.g., Biology), generation quality improves because generator feedback directly informs key updates, creating a tighter retrieval-generation feedback loop.

**Inference time efficiency.** As shown in Figure 3, ERM operates at native retrieval speed (150–280 ms per query), yielding a 50–100× latency reduction compared to HyDE (7–15 seconds) while delivering comparable or superior retrieval quality. Figure 4 further shows that ERM's nDCG@10 improves monotonically as the adaptation budget increases from 0.3 to 0.8, confirming that key evolution accumulates useful signal rather than overfitting.

### 5.2. Analysis of ERM

**Robustness across query expansion methods.** As shown in Figure 6, ERM improves for retrieval across all combinations of query expansion methods and retrievers on Leet-Code, with gains ranging from +12% (Facet with BM25) to +58% (HyDE with BGE-Large). Dense retrievers show the highest gains (+40–58%), confirming that ERM and query expansion are complementary rather than redundant.

**Effect of index methods.** Title-based indexing combined with ERM produces the best results (in Appendix B.8), while full-document embedding yields the weakest baseline. This supports a two-stage approach: titles provide concise, low-noise initialization, while ERM progressively enriches keys based on downstream feedback.

**Robustness on label-disjoint queries.** BRIGHT StackExchange datasets provide a stringent stress test: five datasets

have zero gold-document overlap across queries, meaning key evolution from one query cannot directly benefit another's gold documents. Despite this, ERM yields consistent BM25 gains (+6–47%), while dense retriever performance remains within ±3% of baseline. This confirms that key updates benefit the triggering query without degrading retrieval quality for unrelated queries.

### 5.3. Discussion

ERM's improvements stem from two complementary mechanisms: (1) *Key enrichment for frequently queried documents*: When a document is successfully retrieved and used in generation, ERM updates its key representation to better align with the query patterns that led to successful retrieval. (2) *Disambiguation through specialization*: As document keys become more specialized toward their successful query patterns, the semantic space becomes less crowded, reducing retrieval confusion between similar but distinct documents. This is particularly beneficial for domains with high lexical overlap (e.g., theorem retrieval) where naive embeddings struggle to distinguish between related concepts. A potential drawback of ERM is that its positive feedback loop may reinforce early retrieval biases, though this can be mitigated by increasing the adaptation batch size and using more patient stopping criteria to promote broader exploration.

## 6. Conclusion

We presented Evolving Retrieval Memory (ERM), a training-free framework that bridges query expansion and key expansion by selectively persisting generator-validated evidence into retrieval keys. Through correctness-gated feedback, selective expansion attribution, and progressive key evolution, ERM accumulates validated semantic signals directly into the document index without retraining any model parameters. We further established formal guarantees that ERM updates never decrease retrieval alignment and that evolving keys converge to a stable index. Experiments across diverse retrieval benchmarks, retrievers, and query expansion methods confirm that ERM consistently improves both

retrieval quality and downstream generation accuracy while operating at native retrieval latency.

## Impact Statement

This paper presents Evolving Retrieval Memory (ERM), which improves retrieval-augmented generation by evolving document index keys from query expansion feedback. ERM operates entirely within existing retrieval infrastructure and does not introduce new data collection, user profiling, or content generation capabilities beyond standard RAG pipelines. Its primary societal benefit is improved factual grounding of LLM outputs, which may reduce hallucinations in knowledge-intensive applications and lower the barrier to deploying adaptive RAG systems in resource-constrained settings without retraining.

A potential risk is that the positive feedback loop in key enrichment may amplify existing retrieval biases, as frequently retrieved documents can accumulate stronger keys over time and marginalize initially underrepresented content; practitioners should therefore monitor retrieval coverage to mitigate this effect.

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

# A. Proofs of Theoretical Guarantees

This section provides proofs for the theoretical results stated in Section 4.3. All guarantees are local (per document), non-parametric, and do not rely on optimizing a global retrieval objective.

## A.1. Proof of Proposition 4.2 (Cumulative Consistency)

*Proof.* Define the per-query attribution increment

$$X_t^{(i,j)} := \mathbf{1}_{i,j}(q_t)\, w_{i,j}(q_t)\, \Delta_{i,j}(q_t).$$

Since queries are i.i.d. and keys are fixed, $\{X_t^{(i,j)}\}_{t \geq 1}$ is an i.i.d. sequence of nonnegative random variables.

*Boundedness.* Because $w_{i,j}(q) \in [0,1]$ and similarity gains are bounded for bounded embeddings, $X_t^{(i,j)}$ has finite expectation.

*Consistency.* By the strong law of large numbers,

$$\frac{s_{i,j}^{(T)}}{T} = \frac{1}{T} \sum_{t=1}^{T} X_t^{(i,j)} \xrightarrow{\text{a.s.}} \mathbb{E}[X_1^{(i,j)}] = \mu_{i,j}.$$

*Top-x selection.* Under the strict gap assumption, eventual ordering of $s_{i,j}^{(T)}$ matches that of $\mu_{i,j}$ almost surely. Since the number of expansion units is finite, the top-$x$ selection stabilises. ☐

*Remark* A.1 (Snapshot approximation). Proposition 4.2 assumes fixed keys to ensure i.i.d. attribution increments. In ERM, keys evolve only after batched evolution steps. Between such steps, keys remain unchanged, and the proposition applies to queries processed within each interval. As ERM approaches saturation, evolution steps become increasingly infrequent, improving the accuracy of this snapshot approximation.

## A.2. Proof of Theorem 4.3 (Stability)

*Proof.* **Part (i).** A finite query stream yields finitely many batches, hence finitely many evolution steps. At each step $\|k_i^{(t+1)} - k_i^{(t)}\| \leq xM$, so the total displacement is finite and $\{k_i^{(t)}\}$ is Cauchy in a complete space. The limit equals $k_i^{(0)} \oplus f(\tau_x^*)$ since the final top-$x$ selection determines the augmentation.

**Part (ii).** Finite total displacement implies the sequence is Cauchy and therefore converges. ☐

## A.3. Proof of Corollary 4.4 (Expected Consistency)

*Proof.* By Proposition 4.2, the top-$x$ operator converges a.s. to the indices with the largest $\mu_{i,j}$. Under additive augmentation and inner-product similarity, the gated similarity gain from a set $S$ of expansion units for a future query $q$ is

$$\text{sim}\big(f(q),\ k_i + \textstyle\sum_{j \in S} f(e_j)\big) - \text{sim}\big(f(q),\ k_i\big) = \sum_{j \in S} f(q)^{\top} f(e_j),$$

where the equality uses bilinearity of the inner product. Taking the expectation over $q \sim \mathcal{Q}$ weighted by the attribution indicators and softmax routing gives $\sum_{j \in S} \mu_{i,j}$. This sum is maximised by selecting the $x$ indices with the largest $\mu_{i,j}$, which is exactly $S^*$. ☐

*Remark* A.2 (Scope of applicability). Corollary 4.4 relies on additive similarity structure. It applies exactly to unnormalised dense retrievers and approximately to cosine-similarity models when key norms vary slowly. It does not extend to sparse or late-interaction retrievers, where the result is interpreted as a consistency property of ERM's attribution mechanism rather than a global optimality guarantee.

## A.4. Proof of Proposition 4.5 (Amortized Cost)

*Proof.* Let $D_T$ denote the number of distinct intents observed in $T$ queries, and let $p_r = r^{-\alpha}/H_{m,\alpha}$ be the probability of intent $r$, where $H_{m,\alpha} = \sum_{r=1}^{m} r^{-\alpha}$. Then $\mathbb{E}[D_T] = \sum_{r=1}^{m} \big[1 - (1 - p_r)^T\big]$. Split the sum at the threshold $r^* = \lfloor (T/H_{m,\alpha})^{1/\alpha} \rfloor$:

- *Head intents* ($r \leq r^*$): $Tp_r \geq 1$, so $1 - (1 - p_r)^T \geq 1 - e^{-1}$, contributing $\Omega(r^*) = \Omega(T^{1/\alpha})$.
- *Tail intents* ($r > r^*$): $Tp_r < 1$, so $1 - (1 - p_r)^T \leq Tp_r$. The tail sum is $\sum_{r>r^*} Tp_r = \Theta(T \cdot (r^*)^{1-\alpha}) = \Theta(T^{1/\alpha})$.

Both parts give $\Theta(T^{1/\alpha})$, confirming $\mathbb{E}[D_T] = \Theta(T^{1/\alpha})$. ERM performs expansion at most once per distinct intent, yielding total cost $D_T \cdot C_\phi$ versus $T \cdot C_\phi$ for full QE. The expected ratio is $\Theta(T^{1/\alpha-1}) \to 0$ as $T \to \infty$. □

## B. Experiments

This section provides detailed descriptions of the retrievers, document indexing methods, and query expansion techniques evaluated in our experiments.

### B.1. Dataset

We evaluate on 13 datasets from two benchmarks: BRIGHT (Su et al., 2024) and BEIR (Thakur et al., 2021). BRIGHT provides both retrieval relevance judgments and generation ground truth (gold answers), enabling end-to-end RAG evaluation. BEIR provides only retrieval-level relevance labels without generation ground truth. Dataset statistics are summarized in Table 3.

**BRIGHT – StackExchange Q&A (7 datasets).** These datasets are sourced from StackExchange forums spanning diverse domains: Biology, Earth Science, Economics, Psychology, Robotics, StackOverflow, and Sustainable Living. Each query is a reasoning-intensive question posted by a real user, and the corpus consists of all answers within the corresponding forum. Queries require multi-step reasoning to identify relevant documents, as surface-level lexical overlap between queries and gold documents is typically low. The number of queries ranges from 101 (Psychology, Robotics) to 118 (Earth Science), with corpus sizes from 50K to 122K documents.

**BRIGHT – Coding & Math (4 datasets).** LeetCode contains competitive programming problems paired with solution discussions; Pony consists of queries about the Pony programming language matched to documentation pages; AoPS (Art of Problem Solving) contains math competition problems with theorem-based solutions; and TheoremQA-T pairs mathematical questions with relevant theorems. These datasets are particularly challenging for retrieval, as queries and documents are expressed in different modalities (e.g., a problem statement vs. a theorem or code solution). Corpus sizes vary widely from 7.9K (Pony) to 414K (LeetCode).

**BEIR (2 datasets).** NFCorpus is a medical information retrieval dataset with 323 queries over 3.1K documents, where relevance is graded on a multi-level scale. SciDocs evaluates scientific document retrieval with 1,000 queries over 4K documents. Unlike BRIGHT, these datasets provide only retrieval relevance labels and do not include generation-level ground truth, so we evaluate them on retrieval metrics only.

### B.2. Retrievers

We evaluate nine retrieval models spanning different architectures, including open-source encoders and commercial embedding APIs:

**Open-Source Models.**

- **GTE-base** (Li et al., 2023): General Text Embeddings model from Alibaba DAMO Academy. A 110M parameter encoder trained on large-scale text pairs with contrastive learning. Produces 768-dimensional embeddings optimized for semantic similarity.
- **BGE-base-en-v1.5** (Xiao et al., 2024): BAAI General Embedding model (base version). A 110M parameter BERT-based encoder with instruction-tuned embeddings. Known for strong performance on MTEB benchmark.
- **BGE-large-en-v1.5** (Xiao et al., 2024): Large variant of BGE with 335M parameters and 1024-dimensional embeddings. Offers improved representation capacity at the cost of higher computational requirements.
- **BGE-M3** (Chen et al., 2024a): Multi-Functionality, Multi-Linguality, and Multi-Granularity embedding model. Supports dense, sparse, and multi-vector retrieval in a single model. Particularly effective for cross-lingual and long-document retrieval.

*Table 3.* Dataset statistics. #Q: number of queries; #D: corpus size; #D+: average number of positive (relevant) documents per query; Q.L./D.L.: average query/document length (GPT-2 tokens). Gen. GT indicates whether generation ground truth is available.

| Benchmark | Dataset | #Q | #D | #D+ | Q.L. | D.L. | Gen. GT |
|---|---|---|---|---|---|---|---|
| BRIGHT-SE | Biology | 103 | 57,364 | 3.6 | 83.6 | 115.2 | ✓ |
| | Earth Science | 118 | 122,388 | 7.7 | 132.4 | 113.3 | ✓ |
| | Economics | 103 | 50,221 | 8.0 | 120.2 | 181.5 | ✓ |
| | Psychology | 101 | 52,841 | 7.3 | 118.2 | 149.6 | ✓ |
| | Robotics | 101 | 62,198 | 5.5 | 120.6 | 818.9 | ✓ |
| | StackOverflow | 117 | 101,100 | 7.0 | 704.5 | 478.3 | ✓ |
| | Sust. Living | 108 | 60,732 | 5.6 | 108.0 | 148.5 | ✓ |
| BRIGHT-CM | LeetCode | 142 | 413,932 | 1.8 | 483.1 | 497.5 | ✓ |
| | Pony | 112 | 7,894 | 22.5 | 98.3 | 102.6 | ✓ |
| | AoPS | 111 | 188,177 | 4.7 | 89.0 | 250.5 | ✓ |
| | TheoremQA-T | 206 | 188,177 | 3.2 | 117.1 | 250.5 | ✓ |
| BEIR | NFCorpus | 323 | 3,129 | 38.2 | — | — | ✗ |
| | SciDocs | 1,000 | 4,021 | 4.9 | — | — | ✗ |

- **MiniLM** (all-MiniLM-L6-v2): Distilled 22M parameter model from Microsoft. Optimized for speed with 384-dimensional embeddings. Provides a good balance between efficiency and quality for resource-constrained settings.

- **DistilBERT** (distilbert-base-uncased): A 66M parameter distilled version of BERT. While not specifically trained for retrieval, serves as a baseline for general-purpose language model embeddings.

- **BM25** (sparse): Classic lexical retrieval using Okapi BM25 scoring. Serves as a non-neural baseline that relies on exact term matching and TF-IDF weighting.

**Commercial Embedding APIs.**

- **Cohere Embed v3** (embed-english-v3.0): Cohere's state-of-the-art embedding model producing 1024-dimensional vectors. Trained on diverse web data with compression-aware training for efficient retrieval. Supports input types (search_document, search_query) for asymmetric retrieval.

- **Voyage-2** (voyage-2): Voyage AI's general-purpose embedding model optimized for retrieval tasks. Produces 1024-dimensional embeddings with strong performance on domain-specific benchmarks. Offers good balance between quality and API cost.

### B.3. Document Indexing Methods

We evaluate four document indexing strategies that determine which portion of each document is embedded for retrieval:

- **Full Document (none)**: Embeds the entire document content. Captures comprehensive information but may introduce noise for long documents and exceed model context limits.

- **Title**: Embeds only the document title or heading. Provides concise representation that often captures the main topic. Particularly effective for Q&A forums where titles are written as questions.

- **Abstract**: Embeds the document abstract or summary. Balances comprehensiveness with conciseness. Most applicable to scientific papers and structured documents.

- **Keywords**: Embeds extracted keywords or key phrases. Focuses on the most salient terms, reducing noise from boilerplate content. Keywords are extracted using TF-IDF weighting and domain-specific heuristics.

### B.4. Query Expansion Methods

We evaluate nine query expansion techniques that augment the original query before retrieval. These methods fall into three categories: (1) LLM-based generation methods that use language models to transform or expand queries, (2) neural rewriting methods that use specialized encoder models, and (3) classical IR methods based on lexical analysis.

**LLM-Based Methods.**

- **HyDE** (Hypothetical Document Embeddings) (Gao et al., 2023a): Generates a hypothetical answer to the query using an LLM, then uses that answer as the search query. Bridges the lexical gap between questions and answers by transforming the query into the document space.

---

**HyDE: Hypothetical Document Generation**

```
<task>
You are an expert knowledge synthesizer implementing HyDE (Hypothetical Document
Expansion).  Your task is to generate a comprehensive, detailed hypothetical
document that would perfectly answer the given question.
</task>

<query>
{query}
</query>

<instructions>
First, analyze the question thoroughly in a <think> section:

1.  QUESTION ANALYSIS - What is the core question being asked?
2.  DOMAIN IDENTIFICATION - What field does this belong to?
3.  ANSWER STRUCTURE PLANNING - How should a comprehensive answer be organized?
4.  TERMINOLOGY MAPPING - What technical terms are essential?
5.  SOURCE PREDICTION - What types of documents would contain this information?

After your detailed reasoning, provide a comprehensive hypothetical document
that directly and thoroughly answers the question with extensive background and
context.
</instructions>
```

---

- **Diver** (Long et al., 2025): Diversified query expansion that generates multiple alternative query formulations to capture different aspects of the information need. Aggregates results from diverse query variants to improve recall.

---

**DIVER: Iterative Query Refinement (Round 1)**

```
Given a query and the provided passages (some may be irrelevant), identify helpful
information and write an improved query that incorporates relevant details.

Query:
{query}

Relevant passages:
{passages}

Write an improved query that incorporates key information from the passages:
```

---

- **Decomposition**: Breaks complex queries into simpler sub-queries using chain-of-thought reasoning. Each sub-query retrieves independently, and results are merged. Particularly effective for multi-hop reasoning questions.

```
<task>
You are an expert at analyzing complex questions and breaking them down into
comprehensive, well-researched sub-queries.
</task>

<query>
{query}
</query>

<instructions>
First, analyze the question thoroughly in a <think> section:

1.  COMPLEXITY ANALYSIS - What makes this question complex?
2.  CONCEPT MAPPING - What are the main concepts involved?
3.  INFORMATION NEED BREAKDOWN - What distinct pieces of information are needed?
4.  SUB-QUERY STRATEGY - How should the query be split to maximize coverage?
5.  RETRIEVAL OPTIMIZATION - How can each sub-query be phrased for optimal
matching?

After your detailed reasoning, provide at least 5 comprehensive sub-queries
that are detailed, self-contained, and cover different aspects of the original
question.
</instructions>
```

- **Facet**: Generates multiple facets or aspects of the query topic using an LLM. Each facet retrieves different relevant content, improving recall for broad or ambiguous topics.

Facet: Multi-Aspect Query Generation

```
<task>
You are an expert at semantic facet analysis for information retrieval.  Your task
is to identify multiple semantic dimensions of a query and generate comprehensive
keyword expansions for each facet.
</task>

<query>
{query}
</query>

<instructions>
First, analyze the query thoroughly in a <think> section:

1.  SEMANTIC STRUCTURE ANALYSIS - What are the distinct semantic dimensions?
2.  FACET IDENTIFICATION - What are the main conceptual facets?
3.  TERMINOLOGY MAPPING PER FACET - For each facet, what are the key terms?
4.  RETRIEVAL STRATEGY - How can facet-specific terms improve retrieval diversity?
5.  COVERAGE OPTIMIZATION - Are there gaps in facet coverage?

After your detailed reasoning, provide at least 6 clearly defined semantic facets
with 10-15 relevant keywords each.
</instructions>
```

- **Contrastive**: Generates contrastive descriptions that highlight what the user is NOT looking for. Uses negative examples to help distinguish between similar but different concepts during retrieval.

Contrastive: Positive/Negative Term Generation

```
<task>
You are an expert at contrastive query expansion for precision-focused information
retrieval.  Your task is to generate comprehensive positive and negative expansion
terms.
</task>

<query>
{query}
</query>

<instructions>
First, analyze the query thoroughly in a <think> section:

1.  INTENT ANALYSIS - What would constitute a relevant/irrelevant document?
2.  POSITIVE TERM STRATEGY - What terms MUST appear in relevant documents?
3.  NEGATIVE TERM STRATEGY - What terms indicate a document is OFF-topic?
4.  DISAMBIGUATION ANALYSIS - Are there multiple interpretations?
5.  PRECISION OPTIMIZATION - How can we maximize precision without sacrificing
recall?

After your detailed reasoning, provide:
POSITIVE EXPANSION (30+ terms):  Core concepts, technical vocabulary, related
processes
NEGATIVE EXPANSION (20+ terms):  Off-topic terms, wrong domain indicators,
misleading homonyms
</instructions>
```

- **Custom-Rewriter**: LLM-based query rewriter that comprehensively rewrites and expands queries with detailed reasoning to maximize retrieval effectiveness.

Custom-Rewriter: Comprehensive Query Rewriting

```
<task>
You are an expert query rewriter for information retrieval systems.  Your task is
to comprehensively rewrite and expand the following query to maximize retrieval
effectiveness.
</task>

<query>
{query}
</query>

<instructions>
First, analyze the query thoroughly in a <think> section:

1.  UNDERSTANDING THE QUERY - What is the user really asking for?
2.  DOMAIN ANALYSIS - What field/domain does this query belong to?
3.  DOCUMENT PREDICTION - What types of documents would answer this query?
4.  EXPANSION STRATEGY - What related concepts should be included?
5.  REWRITING APPROACH - How can the query be made more precise?

After your detailed reasoning, provide a comprehensive rewritten query that
includes extensive background context and uses precise technical terminology.
</instructions>
```

**Neural Rewriting Methods.**

- **BGE-Rewriter**: Uses the BGE instruction-following capability to rewrite queries in a retrieval-optimized format. Transforms natural language questions into search-friendly statements without requiring LLM inference.
- **Turbo-Rewriter**: Fast query rewriting using GPT-3.5-Turbo with minimal prompting. Balances quality and latency for

*Table 4.* Naive retrieval: best retriever and index per dataset (393 experiments, no query expansion).

| Dataset | Retriever | Index | NDCG@1 | NDCG@5 | NDCG@10 | Recall@5 | Recall@10 | MRR |
|---|---|---|---|---|---|---|---|---|
| *BEIR Datasets* | | | | | | | | |
| NFCorpus | BGE-base | abstract | 0.288 | 0.155 | 0.125 | 0.026 | 0.031 | 0.351 |
| SciDocs | GTE-base | none | 0.466 | 0.251 | 0.191 | 0.110 | 0.148 | 0.532 |
| *BRIGHT - StackExchange Q&A* | | | | | | | | |
| Biology | GTE-base | title | 0.971 | 0.598 | 0.597 | 0.491 | 0.500 | 0.979 |
| Earth Sci. | GTE-base | title | 0.767 | 0.510 | 0.516 | 0.439 | 0.465 | 0.826 |
| Economics | GTE-base | title | 0.641 | 0.431 | 0.440 | 0.381 | 0.402 | 0.695 |
| Psychology | GTE-base | title | 0.535 | 0.351 | 0.382 | 0.304 | 0.351 | 0.585 |
| Robotics | BGE-M3 | title | 0.455 | 0.350 | 0.361 | 0.342 | 0.369 | 0.543 |
| StackOvflw | GTE-base | abstract | 0.581 | 0.399 | 0.411 | 0.365 | 0.393 | 0.645 |
| Sust. Living | GTE-base | title | 0.750 | 0.500 | 0.511 | 0.430 | 0.465 | 0.808 |
| *BRIGHT - Coding & Math* | | | | | | | | |
| LeetCode | BGE-large | keywords | 0.183 | 0.214 | 0.215 | 0.204 | 0.239 | 0.348 |
| Pony | DistilBERT | none | 0.893 | 0.559 | 0.450 | 0.202 | 0.286 | 0.925 |
| AoPS | GTE-base | keywords | 0.027 | 0.139 | 0.177 | 0.107 | 0.185 | 0.235 |
| TheoremQA-T | MiniLM | abstract | 0.342 | 0.272 | 0.324 | 0.225 | 0.336 | 0.462 |

production settings where full LLM generation is too slow.

## Classical IR Methods.

- **Synonyms**: Expands queries with WordNet synonyms and related terms. A classical IR technique that addresses vocabulary mismatch without neural generation.
- **PRF** (Pseudo-Relevance Feedback): Uses terms from top-k initially retrieved documents to expand the query. Assumes top results are relevant and extracts discriminative terms.

## B.5. Naive Retrieval Analysis

This section provides detailed analysis of naive retrieval experiments, which evaluate different dense retrievers and document indexing strategies without query expansion or dynamic key evolution. These results establish the foundation for comparing our ERM approach. For detailed descriptions of each retriever and index method, see Section B.2 and Section B.3.

## B.6. Experimental Setup

We evaluated 7 dense retrievers across 13 datasets with 4 index methods, totaling 393 naive retrieval experiments. See Section B.2 for retriever details and Section B.3 for index method descriptions.

## B.7. Best Configuration per Dataset

Table 4 presents the best retriever and index configuration for each dataset.

**GTE-base as the dominant retriever.** GTE-base achieves the best performance on 8 out of 13 datasets, spanning both StackExchange Q&A and BEIR domains. This dominance likely stems from GTE's contrastive pre-training on large-scale text pairs, which produces embeddings well-suited for semantic matching across diverse domains. The remaining datasets where GTE does not win, such as Robotics (BGE-M3), NFCorpus (BGE-base), LeetCode (BGE-large), Pony (DistilBERT), and TheoremQA-T (MiniLM), tend to involve either highly specialized vocabularies or document structures that deviate from standard prose, suggesting that domain-specific embedding properties can still outweigh general-purpose representation quality.

**Title indexing preferred for Q&A domains.** StackExchange datasets consistently achieve their best retrieval scores with title-based indexing, yielding 15–40% absolute improvement in NDCG@10 over full-document indexing. This is because

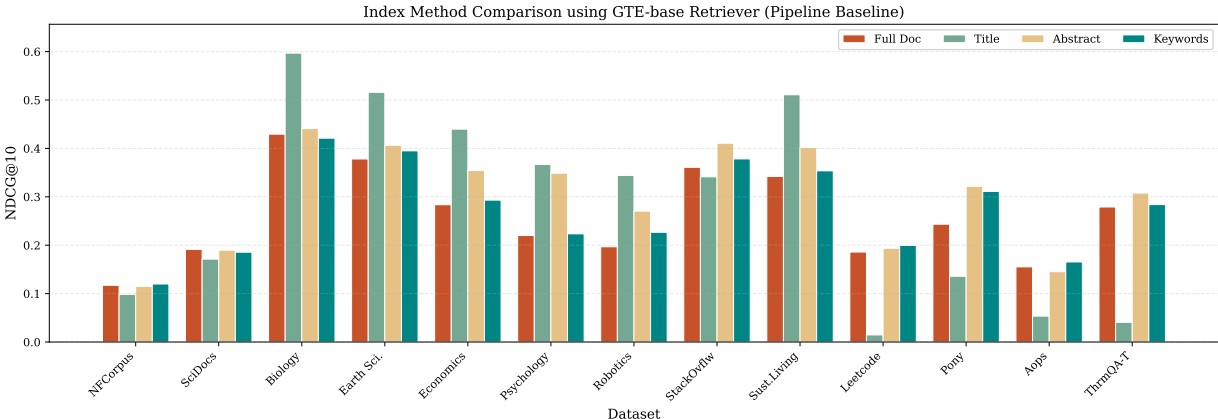

*Figure 5.* Index method comparison using GTE-base retriever across all datasets. Title indexing dominates for StackExchange Q&A domains, while abstract/keywords work better for technical and mathematical content. The consistent advantage of title indexing for Q&A suggests that concise document representations reduce noise in dense retrieval.

StackExchange titles are concise, information-dense summaries that closely mirror the query intent, while full document bodies introduce substantial noise from discussion threads, tangential comments, and code snippets. The effectiveness of title indexing for Q&A content highlights a broader principle: when the query-document matching problem can be reduced to matching against a compact, high-signal representation, dense retrievers benefit from reduced embedding dilution.

**Domain-specific indexing patterns.** Coding and math datasets exhibit different indexing preferences from Q&A domains. LeetCode and AoPS perform best with keyword indexing, where extracted keywords capture the algorithmic concepts and mathematical terms essential for matching. In contrast, BEIR datasets show mixed preferences: NFCorpus favors abstract indexing (capturing medical summaries), while SciDocs works best with no indexing preprocessing. These patterns suggest that the optimal indexing strategy depends on the structural relationship between queries and documents: when queries and documents share explicit technical vocabulary, keyword extraction is effective, while topical but not lexical overlap calls for richer representations.

### B.8. Index Method Comparison

Figure 5 shows the impact of different document indexing strategies using the most-winning retriever (GTE-base) across all datasets, providing a controlled comparison that isolates the effect of document representation from retriever choice.

**Title indexing excels for StackExchange.** Biology, Earth Science, Economics, Psychology, and Sustainable Living all achieve their highest NDCG@10 with title-only indexing, often by significant margins of 0.1–0.2 absolute improvement. This consistent advantage arises because StackExchange question titles are carefully crafted by users to summarize their information need, creating a natural alignment between query semantics and title embeddings. When full document bodies are included, the embedding is diluted by answer text that may discuss tangential aspects or contain formatting artifacts, reducing retrieval precision.

**Abstract indexing for technical content.** StackOverflow and NFCorpus perform better with abstract indexing, suggesting that code-heavy or technical content benefits from richer contextual representations. For StackOverflow, where answers often contain code blocks alongside explanatory text, abstract-level indexing captures the semantic intent behind code solutions. Similarly, NFCorpus medical documents require sufficient context to disambiguate specialized terminology. This finding implies that the optimal document representation granularity depends on the semantic density of the content: sparse, focused documents benefit from compression (titles), while dense, multi-faceted documents benefit from moderate expansion (abstracts).

**Keywords for mathematical domains.** AoPS achieves its best results with keyword indexing, indicating that theorem-based retrieval benefits from explicit keyword extraction. Mathematical queries often reference specific theorem names, problem types, or algebraic structures that serve as strong retrieval signals when isolated from surrounding prose. This aligns with the observation that mathematical reasoning tasks require identifying precise conceptual anchors rather than

broad topical similarity.

**NDCG@1 > NDCG@5 pattern.** A counterintuitive pattern appears across most BRIGHT datasets where NDCG@1 exceeds NDCG@5. This occurs because BRIGHT datasets typically have a small number of relevant documents per query (avg. 3.6 for Biology), so the ideal DCG gain at cutoff 5 or 10 cannot be fully realized. High NDCG@1 indicates strong top-1 precision, while the lower NDCG@5/10 reflects the structural ceiling imposed by having fewer ground-truth relevant documents than the cutoff depth. This metric behavior is important to consider when interpreting retrieval improvements: gains in NDCG@10 are harder to achieve and thus more meaningful for these datasets.

## B.9. Query Expansion Analysis

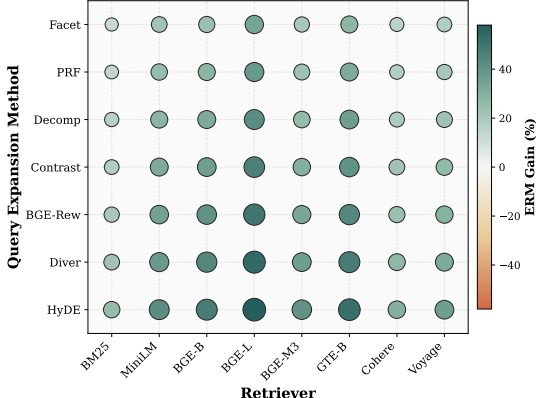

*Figure 6.* **ERM gains across query expansion methods and retrievers on LeetCode.**

**Robustness to different query expansion methods.** As shown in Figure 6, ERM provides consistent improvements over naive retrieval across all combinations of query expansion methods and retrievers on the LeetCode dataset. The bubble sizes and colors indicate the percentage gain achieved by ERM, ranging from +12% (Facet with BM25) to +58% (HyDE with BGE-Large). Dense retrievers (BGE-Large, GTE-Base) show the highest gains (+40–58%), while HyDE and Diver query expansion methods consistently outperform other QE approaches.

Table 5 presents the best query expansion configuration for each dataset. We compare against the naive retrieval baseline (no QE) to measure the impact of different QE methods. QE provides value even when the original queries are already well-formed natural language questions. This indicates that the key evolutionary mechanisms of ERM and query expansion techniques are complementary, not redundant.

**HyDE and Diver as complementary strategies.** HyDE dominates on BEIR and technical content (NFCorpus, SciDocs, Psychology, StackOverflow, LeetCode), while Diver excels on StackExchange Q&A datasets (Earth Science, Economics, Robotics, Sustainable Living). This complementarity arises from their different expansion mechanisms: HyDE generates a hypothetical relevant document and embeds it, which is particularly effective when the retrieval gap stems from modality differences between queries and documents (e.g., problem statements vs. code solutions). Diver, by contrast, generates diverse query reformulations that cover different aspects of the information need, which is beneficial for Q&A domains where relevant documents may address the question from multiple angles.

**Retriever-QE synergy.** GTE remains the most effective retriever even with query expansion active, but the addition of QE enables BGE variants and MiniLM to become competitive on specific domains. Notably, MiniLM with HyDE achieves the best result on LeetCode despite being a smaller model, suggesting that QE can compensate for limited model capacity by transforming queries into a representation space where the retriever is more effective. This finding has practical implications: smaller, faster retrievers paired with QE may achieve comparable performance to larger retrievers without QE, offering a favorable compute-accuracy trade-off.

**Minor regressions on well-structured content.** Biology ($-0.7\%$), Sustainable Living ($-0.5\%$), and Pony ($-0.4\%$) show slight performance drops with QE. These datasets share a common property: their queries and documents already exhibit strong lexical and semantic alignment, so the additional expansion introduces noise rather than useful signal. For Biology and Sustainable Living, the title-indexed documents already provide near-perfect top-1 precision (NDCG@1 of 0.971 and 0.750 respectively), leaving little room for improvement and making even small noise injections harmful. This motivates ERM's selective evolution approach, which conditions updates on verified retrieval improvements rather than applying expansion uniformly.

**Distribution Analysis.** Beyond comparing only the best configurations, we analyze the full distribution of NDCG@1 scores across all retriever–index–QE combinations to understand how query expansion affects retrieval robustness more

*Table 5.* Query expansion results: best QE configuration per dataset (ERM=False). $\Delta(\%)$ shows percentage improvement in NDCG@1 vs naive retrieval baseline.

| Dataset | QE Method | Retriever | Index | NDCG@1 | NDCG@5 | NDCG@10 | Recall@10 | MRR | $\Delta_{\text{NDCG@1}}(\%)$ |
|---|---|---|---|---|---|---|---|---|---|
| *BEIR Datasets* | | | | | | | | | |
| NFCorpus | hyde | BGE-B | abstract | 0.293 | 0.152 | 0.129 | 0.036 | 0.346 | +2.1% |
| SciDocs | hyde | GTE | none | 0.522 | 0.267 | 0.202 | 0.154 | 0.566 | +12.2% |
| *BRIGHT - StackExchange Q&A* | | | | | | | | | |
| Biology | diver | GTE | title | 0.964 | 0.595 | 0.594 | 0.500 | 0.974 | -0.7% |
| Earth Sci. | diver | GTE | title | 0.828 | 0.540 | 0.545 | 0.484 | 0.874 | +8.0% |
| Economics | diver | GTE | title | 0.651 | 0.433 | 0.443 | 0.407 | 0.698 | +1.6% |
| Psychology | hyde | GTE | title | 0.654 | 0.420 | 0.443 | 0.381 | 0.689 | +22.4% |
| Robotics | diver | BGE-M | title | 0.469 | 0.358 | 0.363 | 0.361 | 0.551 | +3.1% |
| StackOvflw | hyde | GTE | keywords | 0.617 | 0.406 | 0.420 | 0.391 | 0.666 | +6.2% |
| Sust. Living | diver | BGE-L | title | 0.770 | 0.502 | 0.505 | 0.447 | 0.815 | +2.7% |
| *BRIGHT - Coding & Math* | | | | | | | | | |
| LeetCode | hyde | Mini | keywords | 0.263 | 0.252 | 0.255 | 0.287 | 0.407 | +43.7% |
| Pony | custom_rewriter | Dist | none | 0.889 | 0.559 | 0.448 | 0.287 | 0.923 | -0.4% |
| AoPS | decomposition | BGE-L | keywords | 0.227 | 0.201 | 0.230 | 0.225 | 0.356 | +740.7% |
| TheoremQA-T | bge_rewriter | GTE | abstract | 0.672 | 0.491 | 0.529 | 0.474 | 0.767 | +96.5% |

broadly. Figure 7 visualizes these distributions as grouped box plots, comparing naive retrieval against query expansion for each dataset. This distributional view complements the per-dataset best-configuration analysis in Table 5 by revealing not only whether the best case improves, but also how the typical case and worst case are affected.

**Consistent improvement on reasoning-intensive datasets.** Biology, Earth Science, Psychology, and AoPS show clear upward shifts in median NDCG@1 with query expansion. The most dramatic improvement appears on AoPS, where the median rises from 0.027 to 0.067, a 148% relative gain. This disproportionate benefit for reasoning-intensive datasets is expected: when the semantic gap between queries and documents is large (as in mathematical problem-to-theorem matching), query expansion provides the intermediate representations needed to bridge that gap. The consistent direction of improvement across these datasets, despite using different optimal QE methods, confirms that the benefit of expansion is a general phenomenon rather than an artifact of a specific technique.

**Reduced variance with query expansion.** Several datasets (Biology, Earth Science, Psychology) exhibit tighter NDCG@1 distributions with QE, as evidenced by shorter interquartile ranges. This variance reduction indicates that QE makes retrieval performance more robust to the choice of retriever and index method. In practical terms, this means that practitioners can achieve more predictable performance with QE even without extensive hyperparameter tuning of the retrieval pipeline, which is an important property for deployment in new domains where the optimal retriever-index combination is unknown a priori.

**Mixed results on BEIR datasets.** NFCorpus and SciDocs show minimal distributional shift between naive and QE-augmented retrieval. These BEIR datasets were designed for standard information retrieval evaluation where queries and documents share substantial lexical overlap, unlike BRIGHT's reasoning-intensive tasks. The limited benefit of QE on BEIR suggests that query expansion is most valuable when there is a fundamental representation gap between queries and documents, rather than when the challenge is merely discriminating among lexically similar candidates.

**Higher performance ceiling with QE.** The upper whiskers for most datasets extend higher with query expansion, indicating that the best QE configurations outperform the best naive retrieval configurations. This is significant because it shows that QE does not merely raise the floor of performance but also pushes the ceiling, creating new opportunities for methods like ERM to build upon. The gap between the QE ceiling and the QE median also suggests room for adaptive selection of QE strategies, precisely the kind of optimization that ERM's key evolution mechanism can exploit.

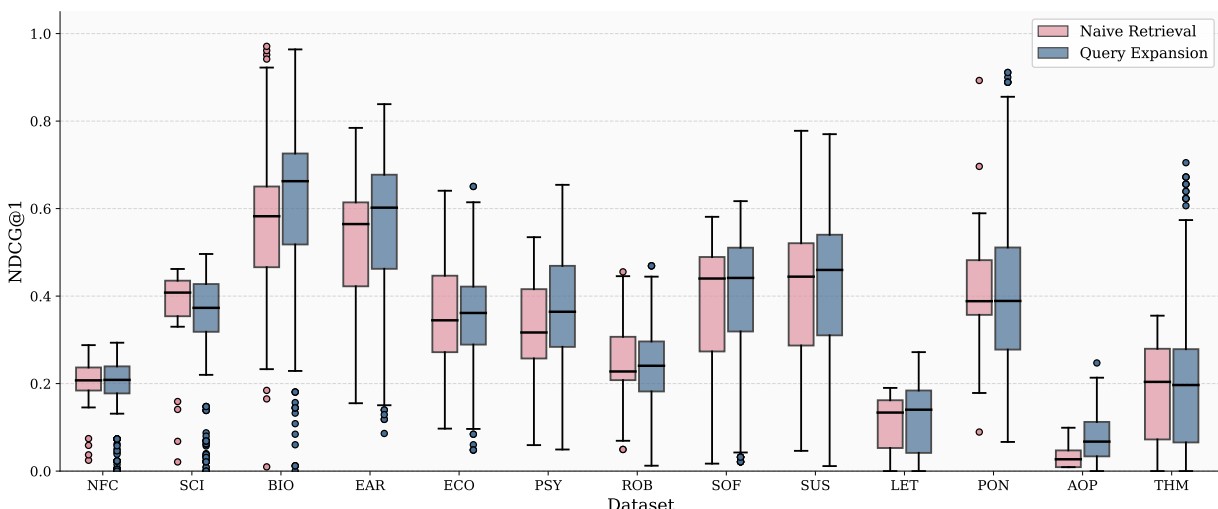

*Figure 7.* Grouped box plot comparing NDCG@1 distributions between naive retrieval and query expansion across all datasets. Each box summarizes all retriever and index configurations tested. Query expansion generally improves median performance and raises the performance ceiling, particularly for complex reasoning datasets like AoPS (+148% median improvement) and Biology (+14% median improvement).

### B.10. Streaming Continual Retrieval

The protocols in the main paper hold the corpus fixed and only vary which queries drive adaptation. To assess ERM under a fully dynamic deployment, where both documents and queries arrive continuously and from shifting domains. We stream all ten BRIGHT domains, introducing them one at a time in alphabetical order ($\mathcal{D}_1$: AoPS, $\mathcal{D}_2$: Biology, ..., $\mathcal{D}_{10}$: Sustainable Living). At each session $t$, a new domain corpus $\mathcal{D}_t$ is added to the index, and its queries are merged into a shuffled query pool, simulating multi-domain traffic in which new and recurring topics arrive simultaneously. In each session, $50\%$ of the pooled queries are randomly sampled and ERM evolves the index *online* for these seen queries; the remaining $50\%$ are deferred to the next session's pool, and the process repeats until all queries are exhausted. Consequently, the first time each domain is introduced, neither its queries nor its documents have been seen during prior updates, so its evaluation measures genuine out-of-distribution (OOD) generalization. To align with production settings, we use Cohere `embed-english-v3` to produce embeddings.

Table 6 reports per-domain retrieval quality at the final session ($t$=10), after the full stream of ten corpora and their interleaved queries has been processed.

*Table 6.* **Streaming continual retrieval on all 10 BRIGHT domains.** NDCG@10 at the final session ($t$=10), comparing native retrieval (Baseline) against ERM with online key evolution. $\Delta$ is the absolute gain (ERM $-$ Baseline) in NDCG@10 points.

| Domain | Baseline | ERM | $\Delta$NDCG@10 |
|---|---|---|---|
| AoPS | 0.142 | 0.155 | +1.3 |
| Biology | 0.415 | 0.420 | +0.5 |
| Earth Science | 0.247 | 0.253 | +0.6 |
| Economics | 0.238 | 0.250 | +1.2 |
| LeetCode | 0.193 | 0.239 | +4.6 |
| Pony | 0.031 | 0.102 | +7.1 |
| Psychology | 0.095 | 0.112 | +1.7 |
| Robotics | 0.150 | 0.150 | 0.0 |
| StackOverflow | 0.203 | 0.229 | +2.6 |
| Sust. Living | 0.216 | 0.229 | +1.3 |

**No catastrophic forgetting.** At the final session, every one of the nine previously introduced domains retains a non-negative $\Delta$NDCG@10, with an average improvement of $+2.2$ points over the baseline. Across the full stream, the worst single-domain degradation at *any* session never exceeds $0.8$ NDCG@10 points. Evolving keys with new-domain signals therefore does not erode retrieval quality on earlier domains; cross-domain memory accumulation is beneficial rather than harmful.

**Positive cross-domain transfer.**    Several domains improve well beyond the noise floor as unrelated corpora and queries continue to stream in: Pony (+7.1), LeetCode (+4.6), StackOverflow (+2.6), Psychology (+1.7), and both AoPS and Sustainable Living (+1.3). Because ERM aggregates expansion signals through selective attribution rather than memorizing query–document pairs, the evolved keys capture shared semantic structure that transfers across topic boundaries.

**Self-correcting behavior.**    The benefit on a freshly introduced domain is not always immediate. Biology, for instance, shows a small transient drop of $-0.8$ points when first added at $\mathcal{D}_2$, but recovers to $+0.5$ by the final session as more cross-domain evidence accumulates. This indicates that ERM tolerates an initial OOD shock and converges to a net-positive state rather than compounding early errors.

Together, these results show that ERM remains stable under simultaneous document and query shift, while enabling positive cross-domain transfer in a fully dynamic, non-i.i.d. environment, directly addressing the streaming and out-of-distribution scenarios that static holdout protocols cannot capture.

