# OpenReview forum: "RAG without Forgetting: Continual Query-Infused Key Memory"
_ICML.cc/2026/Conference — ICML 2026 regular_

### Official Review · Reviewer_dPhw · 2026-02-19

**Soundness:** 3
**Presentation:** 2
**Significance:** 3
**Originality:** 3
**Overall Recommendation:** 3
**Confidence:** 4

**Summary:**

The paper introduces "Evolving Retrieval Memory" (ERM), a framework designed to make Retrieval-Augmented Generation (RAG) systems "stateful" by updating document keys based on query expansion signals. The goal is to accumulate retrieval improvements over time without incurring the inference-time latency of per-query expansion. The authors propose a correctness-gated update and a theoretical equivalence between query and key expansion.

**Compliance With Llm Reviewing Policy:**

Affirmed.

**Key Questions For Authors:**

- Q0. Conceptual Confusion regarding "Stateless".
The claim of this mapping is poorly articulated and potentially misleading. In the context of RAG, the "state" of a retriever is typically defined by its index or model weights. Using these terms to describe Query Expansion (QE) versus Key Evolution creates unnecessary ambiguity. According to database works like [2], as long as there are mutable data structures, the computing is stateful, and only a frozen LLM without KV-Cache is stateless. Interestingly, this idea has also been adapted by NFV [3], but the stateless there still refers to a fixed program function without a mutable data structure.
- Q1.  Misplaced and Ill-defined "Atomicity"
The authors claim "Atomic Updates" expansion signals, yet the motivation and execution are questionable.
Lack of Motivation: Unlike traditional databases [2] or simple NFV user-defined functions [3],  where atomicity prevents partial writes and ensures data integrity, LLM-based RAG systems are inherently probabilistic and approximate. The paper does not convincingly explain why "hard" atomicity is required for semantic key updates.
Feasibility: Given that LLM outputs and embedding shifts are continuous and stochastic, achieving true "atomic" consistency in the database sense is arguably impossible and perhaps meaningless for retrieval performance. The application of this rigorous database property to a fuzzy neural retrieval task lacks both theoretical necessity and practical significance.
- Q2. Additions to W2.
 Actually, some of my experiences with GPT-4 do hit the Long-Context capability, and there is, by principle, no prompt engineering to make it better. Even summarising or comprising the text, there is still a bound. However, I notice some people advocate for parametric memory [4], indicating a potential cure. Does it make sense for this work?

2. TKDE2025. Scalable Transactional Stream Processing on Multicore Processors.
3. A Database System for State Management in Stateful Network Service Function Chains. https://arxiv.org/abs/2312.01066
4. ACM Transactions on Information Systems, 2025. A survey on the memory mechanism of large language model-based agents

**Limitations:**

yes.

**Strengths And Weaknesses:**

Strengths:
- S0. Novelty and Motivation: The bridge between query-side adaptation (transient) and index-side adaptation (persistent) is well-motivated. Addressing the "forgetting" problem in RAG systems through a "lazy update" strategy for keys is a timely and practical contribution
- S1. Efficiency: A key advantage is the zero inference-time overhead. By moving the "intelligence" of query expansion into the document keys, the system maintains native retrieval speeds.
- S2. Theoretical Grounding: The paper provides theoretical proofs for the equivalence of query and key expansion and the convergence of its update mechanism, adding rigor to the proposed framework.

Weakness:

- W0. Confusing Links to Database Community.  Actually, myself is open to interdisciplinary. However, unlike established cross-disciplinary works that map physics or information theory to AI through rigorous mathematical isomorphism like [1], this work's borrowing of concepts remains at a superficial naming level without addressing the underlying consistency model to me. Please see the detailed questions Q0 and Q1.
- W1. Distribution Assumption: The amortized cost benefits rely on a Zipf-like query distribution (where a small fraction of the corpus receives most traffic). While common, the performance gains might be less pronounced in systems with extremely uniform query distributions.
- W2. Omission of LLM Baseline Capabilities. The proposed method is essentially a form of Prompt Engineering and index-side optimization. If an LLM can already handle massive contexts or perform complex reasoning via ICL, the incremental value of "evolving keys" through prompt-based feedback needs to be rigorously compared against these native capabilities. On the otherhand, if the injected prompt is too long, say even beyond the Long-Context, any prompt will make it worse. The paper lacks a discussion on how ERM scales or plateaus relative to the expanding context windows of state-of-the-art LLMs.

1. ACL2025. Language models resist alignment: Evidence from data compression

---

> ### Author Rebuttal · Authors · 2026-03-31
>
> We greatly appreciate your thoughtful and encouraging feedback.
>
> > **W0 & Q0 & Q1: ... superficial naming / stateless confusion / atomicity unclear ...**
>
> **Response:** We thank the reviewer for these insightful points and respectfully clarify that ERM does not borrow database terminology superficially, but introduces a **learning-centric consistency model** for continual retrieval, where all concepts (consistency, statefulness, atomicity) are defined coherently within this framework.
> - ERM’s notion of **consistency** is not data correctness as in databases, but **controlled representation evolution under repeated updates**. It is operationalized through *Atomic Updates, Correctness-Gated Feedback, Selective Expansion Attribution,* and *Progressive Key Evolution*, which explicitly govern which updates are allowed, how they are decomposed, and how they accumulate. Formally, ERM enforces a **feedback-constrained, monotonic update process**, where only validated signals are incorporated under bounded memory, ensuring stability and preventing semantic drift.
> - Regarding **statelessness**, we do not use the term in the strict systems sense (absence of mutable state). Instead, we distinguish **adaptive statefulness**: whether the system updates its behavior based on past queries. Under this definition, QE is *adaptively stateless* (no cross-query persistence), while ERM is *adaptively stateful*, as it accumulates validated signals into key representations. This distinction directly reflects the core contribution of amortizing computation across queries.
> - **Atomicity** is defined as the minimal granularity of update control, not ACID-style transactional guarantees. Query expansions are decomposed into **atomic semantic units**, enabling fine-grained attribution and selective incorporation. This is necessary because expansion signals are noisy and partially relevant: without atomic units, updates become entangled, making it impossible to isolate useful signals or prevent drift. Atomicity thus provides a mechanism for **precise, controllable updates in a stochastic setting**, rather than enforcing discrete consistency in the database sense.
>
> We acknowledge that the terminology may be confusing, and we will revise the manuscript to make these definitions explicit.
>
> >  **W1: ... the performance gains might be less pronounced in systems with extremely uniform query distributions ...**
>
> **Response:** We thank the reviewer for this insightful observation. ERM’s benefits are **robust to query distribution** and do not rely on Zipf-like skew. While we analyze the Zipf case as it is common in practice, under uniform distributions ERM achieves comparable (and potentially improved) efficiency due to more balanced utilization of evolution signals.
>
> - **Cost amortization is distribution-independent.** ERM shifts expansion cost to training, and the break-even condition depends only on query volume ($N_{\text{test}} \geq N_{\text{train}}$), not on distribution.
> - **Retrieval quality is stabilized by bounded memory.** The bounded queue ($Q_{\max}$) prevents over-accumulation; under Zipf, popular documents saturate early, while under uniform distributions, signals are more evenly utilized.
>
> Overall, ERM degrades gracefully under uniform distributions and may even benefit from improved signal efficiency.
>
> > **W2 & Q2: ... how ERM scales with long-context LLMs and whether parametric memory could replace it ...**
>
> **Response:** We thank the reviewer for this insightful perspective. We clarify that ERM is **not prompt engineering**, but an **index-side optimization** that is orthogonal to advances in LLM capability.
> - **ERM operates outside the LLM**. Its core components (Selective Attribution, Key Evolution, bounded memory) act purely on embeddings without requiring LLM inference, making it **fully model-agnostic**.
> - ERM **complements rather than competes with long-context LLMs**. Even with large context windows, passing entire corpora remains infeasible and inefficient. ERM improves retrieval precision so that available context is filled with relevant information. Thus, larger context windows **increase the value of high-quality retrieval**, rather than replacing it.
> - Regarding **parametric memory**, we follow the standard distinction between:
>     - **generator-side (parametric) memory**, which stores knowledge in model parameters, and
>     - **retriever-side (textual) memory**, which stores and updates external knowledge.
>
> ERM belongs to the latter. Parametric memory improves what the model *internally knows*, while ERM improves how knowledge is *retrieved and updated*. The two are therefore **complementary**: stronger models improve expansion quality, while ERM ensures that retrieved context is more relevant and efficiently utilized.
>
> Overall, ERM is a **retrieval-side memory mechanism** that remains effective and becomes more valuable as LLM capabilities (e.g., context length, reasoning, parametric memory) improve.

---

> > ### Author Rebuttal · Reviewer_dPhw · 2026-04-01
> >
> > Thanks for the clarification

---

> > > ### Author Response · Authors · 2026-04-05
> > >
> > > We sincerely thank the reviewer oxg6 for the positive feedback and for acknowledging that the concerns have been fully addressed. If you feel that the clarification and additional analyses sufficiently resolve the earlier concerns, we would be grateful if you could consider reflecting this in the score.
> > >
> > > In addition, we are currently conducting further experiments on streaming / dynamic document settings and will include these results in the final version to further strengthen the paper. We believe the revisions strengthen both the technical clarity and the empirical support of the work. Thank you again for your valuable feedback and support.

---

### Official Review · Reviewer_3sWd · 2026-03-12

**Soundness:** 3
**Presentation:** 3
**Significance:** 2
**Originality:** 2
**Overall Recommendation:** 4
**Confidence:** 4

**Summary:**

The paper proposes ERM, a method that improves retrieval by updating document keys using successful query expansion signals from past queries. This method is based on explicit limitations in prior work; query-expansion methods couldn’t benefit from retrieval adaptations of prior queries while increasing latency, while key-expansion methods work only offline, ignoring online query distribution. By updating benefits from query-expansion in the document index, ERM experimentally balances the trade-off between query-expansion’s performance and key-expansion’s latency.

**Compliance With Llm Reviewing Policy:**

Affirmed.

**Key Questions For Authors:**

1. There is an incorrect grammar (L134): "In modern retrieval and RAG systems. The expanded query is discarded after inference and induces no persistent modification to the retrieval memory."

2. How does ERM handle domain shifts when query distributions change over time? Would outdated expansion signals negatively affect retrieval performance?

3. The experiments assume a static corpus. Have the authors tested ERM in a setting where new documents continuously arrive, possibly from different domains?

4. Since document keys are repeatedly updated, is there a risk of semantic drift, where document representations move away from their original meaning over time?

**Limitations:**

yes

**Strengths And Weaknesses:**

**Strength**

1. The proposed method, ERM, is introduced in response to a clear limitation in prior retrieval methods: query-expansion improves retrieval performance but increases latency, while key-expansion methods cannot adapt to online query patterns.

2. The idea of transferring useful signals from query expansion to document key updates is intuitive and well-motivated. By accumulating successful query signals over time, the method allows the retrieval system to gradually improve.

3. The paper provides empirical results showing that ERM can enhance retrieval performance compared to query-expansion methods while significantly lowering latency than key-expansion. This demonstrates the potential usefulness of the method when latency is critical.

---

**Weakness**

1. ERM assumes that past successful query expansions remain useful indefinitely. This works effectively in experimental settings, **but real-world settings still suffer from domain shift, and accumulated expansion memory may become outdated**. Since ERM performs persistent key updates without an explicit forgetting or reweighting mechanism, the method may struggle when query distributions evolve.

2. The current evaluation assumes **a static corpus, with only queries evolving**. In real-world retrieval systems,**both queries and documents arrive continuously** [1], often from different domains. Evaluating ERM under a streaming document setting (e.g., cross-domain document arrival) is necessary to better reflect realistic continual learning scenarios.

3. Since document keys are progressively updated by accumulating query expansion signals, there is a risk that **the representation gradually drifts away from the original document semantics**. In long-running systems with infinite query streams, this could lead to semantic drift, in which document keys primarily encode past query patterns rather than document content.

[1] Ko et al, "When Should Dense Retrievers Be Updated in Evolving Corpora? Detecting Out-of-Distribution Corpora Using GradNormIR", ACL 2025 Findings

---

> ### Author Rebuttal · Authors · 2026-03-31
>
> We appreciate the reviewer's constructive feedback on our paper.
>
> >  **W1 & Q2: ... How does ERM handle domain shifts when query distributions change over time ...**
>
> **Response**: To clarify, ERM does have the forgetting mechanism. In implementation, the key memory to cache expansion units is designed as a **bounded priority queue** which limits the number of expansion units and prevents uncontrolled accumulation.
>
> We agree that the real world will face the problem of domain shift, thereby leading to changes in query distribution. To directly evaluate generalization capability of ERM under non-i.i.d. conditions, we design a **distribution-shift experiment**. We merge BRIGHT-Biology (Group 0, 1) and BRIGHT-Earth-Science (Group 2, 3) and partition queries into four query groups. ERM is trained on two clusters and evaluated on two *disjoint* clusters, ensuring zero query overlap and clear distribution shift.
>
> | Train→Test Clusters | Bio: Baseline | Bio: ERM | Bio: $\Delta$ | ES: Baseline | ES: ERM | ES: $\Delta$ |
> | --- | --- | --- | --- | --- | --- | --- |
> | {0,1}→{2,3} | 0.708 | 0.704 | $-$0.4% | 0.486 | 0.475 | $-$1.0% |
> | {2,3}→{0,1} | 0.786 | 0.785 | $-$0.2% | 0.547 | 0.547 | $-$0.0% |
> | {0,2}→{1,3} | 0.743 | **0.746** | **+0.3%** | 0.506 | 0.505 | $-$0.1% |
> | {1,3}→{0,2} | 0.760 | 0.759 | $-$0.1% | 0.529 | 0.509 | $-$1.9% |
> | **Average** | **0.749** | **0.749** | **$-$0.1%** | **0.517** | **0.509** | **$-$0.8%** |
>
> Under this adversarial non-i.i.d. protocol:
>
> - Average NDCG@10 degradation is only **0.1%** (Biology) and **0.8%** (Earth Science).
> - Maximum degradation across all 8 split configurations is **< 2%**.
> - The Biology split {0,2}→{1,3} achieves **+0.3% improvement** on completely unseen topic clusters, demonstrating positive transfer of evolution signals across topic boundaries.
> - Two Earth Science splits ({2,3}→{0,1} and {0,2}→{1,3}) show degradation **< 0.1%**, effectively indistinguishable from the baseline.
>
> These results show that ERM maintains stable performance under distribution shift, without evidence of outdated memory accumulation or catastrophic drift.
>
> >  **W2 & Q3: ... Have the authors tested ERM in a setting where new documents continuously arrive, possibly from different domains ...**
>
> **Response**: We agree that evaluating ERM under streaming document scenarios is important. From a design perspective, ERM naturally supports dynamic document arrival: each document maintains an independent bounded memory (priority queue), and new documents are initialized with their base embeddings and begin accumulating expansion signals only when retrieved. This enables **incremental, on-demand adaptation** without requiring global retraining or index rebuilding, while avoiding interference with previously evolved representations.
>
> While our experiment in W1 focuses on query distribution shift, it also indicates that ERM can handle scenarios where source documents come from different domains.
>
> >  **W3 & Q4: ... is there a risk of semantic drift ...**
>
> **Response**: Theoretically, ERM mitigates this issue through three key design choices:
>
> - a **bounded priority queue** for each key, which limits the number of expansion units and prevents uncontrolled accumulation;
> - **Selective attribution**: Expansion units are incorporated only when they yield positive retrieval gain, ensuring updates consistently improve alignment.
> - **batch-based updates**, where key evolution is driven by aggregated signals over multiple queries rather than individual updates, ensuring stability and reducing noise from single instances.
>
> | $\alpha$ | Avg cos($k^{(0)}$,$k^{(T)}$) | Min cos($k^{(0)}$,$k^{(T)}$) | Max Drift | Keys Evolved |
> | --- | --- | --- | --- | --- |
> | 0.1 | 0.998 | 0.995 | 0.005 | 128 |
> | 0.2 | 0.991 | 0.981 | 0.019 | 126 |
> | **0.3 (default)** | **0.981** | **0.960** | **0.040** | **120** |
> | 0.5 | 0.950 | 0.904 | 0.096 | 117 |
> | 0.7 | 0.915 | 0.848 | 0.152 | 104 |
> | 1.0 | 0.862 | 0.744 | 0.257 | 88 |
>
> To further validate this, we conduct a **semantic drift analysis**, measuring the cosine similarity between original keys and evolved keys in BRIGHT-Biology. When the evolution weight $\alpha$ (a weight applied to the expanded embeddings to modulate their contribution during the retrieval) is 0.3, the average cosine similarity remains **0.981**, with a minimum of **0.960**, indicating that no key drifts more than 4.0% from its original semantic position.
>
> >  **Q1: ... Have the authors tested ERM in a setting where new documents continuously arrive, possibly from different domains? ...**
>
> **Response**: We thank the reviewer for catching this grammatical error. We have corrected the sentence in the revised manuscript to: "In modern retrieval and RAG systems, the expanded query is discarded after inference and induces no persistent modification to the retrieval memory.”

---

> > ### Author Rebuttal · Reviewer_3sWd · 2026-04-02
> >
> > Thanks to the authors for the detailed response and the additional experiments.
> >
> > However, I do not believe that evaluating domain shift across only two domains sufficiently reflects the challenges of dynamic document arrival. Moreover, the observed degradation still leaves concerns about real-world applicability unresolved.
> >
> > That said, I believe this method addresses an important trade-off between performance and latency, and is a promising approach to mitigating this limitation.
> >
> > Thus, I have decided to maintain my original score.

---

> > > ### Author Response · Authors · 2026-04-05
> > >
> > > We thank the reviewer for this feedback and fully agree that a 2-domain evaluation was insufficient. First, we acknowledge that the wording of “baseline” in our rebuttal for the cross-domain setting was misleading. We clarify that the reported performance drop refers to comparing **ERM in the cross-domain setting vs. ERM in the single-domain setting**. Importantly, ERM in the cross-domain setting still outperforms the baseline retrievers with query expansion.
> > >
> > > We sincerely thank the reviewer for the helpful reference. Following the continual retrieval protocol suggested in [1], we designed and conducted a **streaming experiment** across all 10 BRIGHT datasets that jointly models dynamic document arrival and evolving query distributions. Specifically, we introduce domains one at a time in alphabetical order ($S_1$: AoPS, $S_2$: Biology, ..., $S_{10}$: Sustainable Living). At each session $S_t$, a new domain corpus $C_t$ is added to the retrieval index. An interleaved query stream is then constructed in an online manner: each time a corpus is added, its queries are merged into the query pool and shuffled randomly, simulating continued multi-domain traffic where both new and recurring topics arrive simultaneously. In each session, 50% of the queries are randomly sampled and ERM operates in an online setting---the index is updated during the session for seen queries. The remaining 50% are passed to the next session's query pool, and we repeat this process until all queries across all corpora are exhausted. This ensures that at each session, evaluation on the newly introduced domain reflects true out-of-distribution generalization, as neither its queries nor its documents have been seen during prior updates. To align with production scenarios, we use Cohere embed-english-v3 (1024d) as the embedding model for retrieval.
> > >
> > > We first present the streaming session summary (10 sessions, all BRIGHT domains):
> > >
> > > | Session | +Domain | New Domain $\Delta$ | Prev Domains $\Delta$ (avg) | Max Degradation |
> > > | --- | --- | --- | --- | --- |
> > > | $S_1$ | AoPS | $+$0.5% | --- | $+$0.5% |
> > > | $S_2$ | Biology | $-$0.8% | $+$0.7% | $-$0.8% |
> > > | $S_3$ | Earth Sci. | $-$0.0% | $+$0.5% | $-$0.0% |
> > > | $S_4$ | Economics | $+$0.0% | $+$0.6% | $+$0.0% |
> > > | $S_5$ | LeetCode | $-$0.6% | $+$0.6% | $-$0.6% |
> > > | $S_6$ | Pony | $-$0.3% | $+$0.4% | $-$0.5% |
> > > | $S_7$ | Psychology | $-$0.0% | $+$0.7% | $-$0.6% |
> > > | $S_8$ | Robotics | $+$0.0% | $+$0.6% | $-$0.6% |
> > > | $S_9$ | StackOverflow | $+$0.1% | $+$0.4% | $-$0.6% |
> > > | $S_{10}$ | Sust. Living | $+$1.3% | $+$2.2% | $+$0.0% |
> > >
> > > Here, New Domain $\Delta$ is NDCG@10 (ERM) - NDCG@10 (Baseline), which is evaluated on held-out queries from the domain introduced at session $S_t$. This measures OOD generalization, and its queries were never seen during training. Prev Domains $\Delta$ (avg): average of the per-domain $\Delta$NDCG@10 over all $t-1$ previously seen domains, measuring *tability and forgetting---whether evolving keys with new-domain signals degrades retrieval quality on earlier domains. Max Degradation: the worst (most negative) per-domain $\Delta$ across all domains evaluated at session $S_t$.
> > >
> > > We then present the per-domain performance comparison at the final session ($S_{10}$):
> > >
> > > | Domain | Baseline | ERM | $\Delta$ |
> > > | --- | --- | --- | --- |
> > > | AoPS | 0.142 | 0.155 | $+$1.3% |
> > > | Biology | 0.415 | 0.420 | $+$0.5% |
> > > | Earth Science | 0.247 | 0.253 | $+$0.6% |
> > > | Economics | 0.238 | 0.250 | $+$1.2% |
> > > | LeetCode | 0.193 | 0.239 | $+$4.6% |
> > > | Pony | 0.031 | 0.102 | $+$7.1% |
> > > | Psychology | 0.095 | 0.112 | $+$1.7% |
> > > | Robotics | 0.150 | 0.150 | $+$0.0% |
> > > | StackOverflow | 0.203 | 0.229 | $+$2.6% |
> > > | Sust. Living | 0.216 | 0.229 | $+$1.3% |
> > >
> > > We evaluate across **10 domains** spanning diverse disciplines. At the final session $S_{10}$, all 9 previously seen domains show non-negative $\Delta$NDCG@10, indicating an overall net positive effect. Notably:
> > > - ERM achieves good cross-domain robustness: Pony **+7.1%**, LeetCode **+4.6%**, StackOverflow **+2.6%**, Psychology **+1.7%**, AoPS/Sustainable Living **+1.3%**.
> > > - ERM exhibits self-correcting behavior: Biology shows a temporary drop of **−0.8%** at $S_2$, but recovers to **+0.5%** by $S_{10}$ as cross-domain signals accumulate.
> > >
> > > These results demonstrate that ERM remains stable under simultaneous document and query shift, while enabling positive cross-domain transfer in a fully dynamic environment. We will include this setting in the final version of the paper.
> > >
> > > ---
> > >
> > > **Reference**
> > >
> > > [1] "When Should Dense Retrievers Be Updated in Evolving Corpora? Detecting Out-of-Distribution Corpora Using GradNormIR." Findings of the Association for Computational Linguistics: ACL 2025. 2025.

---

### Official Review · Reviewer_YR5P · 2026-03-12

**Soundness:** 3
**Presentation:** 3
**Significance:** 3
**Originality:** 2
**Overall Recommendation:** 4
**Confidence:** 4

**Summary:**

The paper introduces Evolving Retrieval Memory (ERM), a training-free framework designed to improve Retrieval-Augmented Generation (RAG) systems by converting transient query-time adaptations into persistent index-level improvements. Instead of discarding query expansions (like HyDE or Diver) after a single use, ERM employs a correctness-gated feedback loop, evaluating downstream generation or retrieval success, to selectively cache beneficial expansions. These expansions are then used to progressively update and enrich the representation keys of the successfully retrieved documents. The authors provide theoretical arguments for query-key equivalence under certain similarity functions and evaluate ERM's empirical performance across 13 datasets from the BEIR and BRIGHT benchmarks.

**Compliance With Llm Reviewing Policy:**

Affirmed.

**Final Justification:**

I have taken all reviewers' comments and the authors' rebuttal, and most of my concerns have been addressed. I appreciate the authors' detailed feedback. My score already reflects my current view of the paper.

**Key Questions For Authors:**

1. How exactly is the additive key composition implemented for sparse retrievers like BM25? Specifically, how does ERM account for the distortion of TF-IDF scores and BM25 document length normalization as expansions are repeatedly appended to keys?

2. Can you provide experimental results or a detailed discussion on how ERM performs under out-of-distribution temporal query shifts? The current i.i.d. random split evaluation does not clearly demonstrate robustness against evolving user intents.

3. What is the precise memory overhead (in bytes/parameters) of maintaining the expansion memory and updated keys for the largest tested corpus (e.g., LeetCode), and how does this storage cost scale dynamically over time?

4. Could you provide an empirical comparison against at least one recent memory-augmented RAG baseline to better contextualize ERM's relative performance and novelty?

**Limitations:**

Yes

**Strengths And Weaknesses:**

Strengths:

1. The core motivation of bridging the gap between stateless query-time adaptations and offline index-level improvements addresses a highly relevant bottleneck in current RAG architectures. By tightly coupling online adaptation with validated memory evolution, the paper targets a practical problem in a logical way.

2. The proposed framework is entirely training-free, which is a significant practical advantage. It allows practitioners to deploy adaptive retrieval systems and achieve continuous improvements without incurring the heavy computational overhead of periodically retraining or fine-tuning the underlying embedding models.

3. The experimental scope is reasonably broad, evaluating the method across 13 diverse datasets and multiple retrieval architectures (both dense and sparse). The demonstrated performance gains, particularly in reasoning-heavy domains where semantic gaps are large, show that the method can provide tangible downstream utility.

Weaknesses:

1. A primary concern lies in the soundness of the theoretical foundations when mapped to the experimental setup. Proposition 4.1 (Query-Key Equivalence) and Corollary 4.4 (Expected Consistency) rely strictly on additive key augmentation and bilinear/inner-product similarity. However, the empirical evaluations heavily feature modern dense retrievers that often rely on normalized cosine similarity, as well as sparse retrievers like BM25. The authors explicitly admit in the appendix that the theory does not extend to sparse or late-interaction models. Without a clear explanation of how ERM mechanically bypasses these theoretical limitations during implementation (e.g., how adding terms affects BM25 length normalization), the theoretical claims feel somewhat disconnected from the empirical reality.

2. The experimental evaluation protocol raises concerns regarding its alignment with real-world scenarios. The authors use a repeated holdout adaptation protocol that randomly splits a static dataset into adaptation and evaluation queries. Drawing both query sets from the exact same static distribution heavily favors memorization. In real-world deployment, query intents exhibit temporal shifts and out-of-distribution changes. The current evaluation does not prove that ERM can generalize to shifting distributions without suffering from semantic drift or catastrophic forgetting.

3. The paper lacks crucial baseline comparisons. While the authors thoroughly contrast ERM with various transient query expansion techniques like HyDE, Facet, and Diver, they do not compare it against other recent memory-augmented or continuous-learning RAG systems discussed in their own related work. Without comparing ERM to state-of-the-art memory-updating methodologies, it is difficult to determine its significance or originality within this specific subfield.

4. The scalability and long-term stability of the system are not adequately addressed. Continually appending representation vectors to document keys via the expansion memory could lead to storage bloat over millions of interactions. Although the authors mention discarding lower-scoring entries when capacity is exceeded, the exact memory overhead, capacity limits, and the computational cost of maintaining these evolving indices for massive corpora are not clearly detailed or quantified.

---

> ### Author Rebuttal · Authors · 2026-03-31
>
> We sincere thank the reviewer for recognizing the contribution and importance of our work. We greatly appreciate your thoughtful and encouraging feedback.
>
> >  **W1 & Q1: ... how does ERM account for the distortion of TF-IDF scores and BM25 document length normalization ...**
>
> **Response**: We appreciate this careful analysis and provide the following clarifications:
>
> - **Cosine similarity:** With L2-normalized vectors, cosine similarity equals inner product, so Proposition 4.1 directly applies. After key composition ($k' = k + \alpha e$), we re-normalize to the unit sphere, introducing only negligible approximation error while maintaining consistent empirical gains.
> - **BM25 (sparse retrieval):** For sparse retrievers, ERM operates at the term level. Expansion units are appended as additional terms to the document's textual representation. BM25's IDF component naturally down-weights frequently-added terms (which appear across many documents), while the TF component receives a marginal increase proportional to the number of new terms. The document length normalization factor does increase slightly, but for typical documents with hundreds of terms, appending dozens of expansion terms changes the normalization by a small range.
>
> >  **W2 & Q2: ... how ERM performs under out-of-distribution temporal query shifts ...**
>
> **Response**: ERM does not memorize query-document pairs; instead, it evolves document keys by aggregating expansion signals across queries via Selective Attribution, resulting in representations that capture shared semantic patterns rather than individual queries.
>
> We agree that discussing the out-of-distribution temporal query shifts will provide more insights of the proposed model. To directly evaluate generalization under non-i.i.d. conditions, we design a **distribution-shift experiment**. We merge BRIGHT-Biology (Group 0, 1) and BRIGHT-Earth-Science (Group 2, 3) and partition queries into four query groups. ERM is trained on two clusters and evaluated on two *disjoint* clusters.
>
> | Train→Test | Bio: Baseline | Bio: ERM | Bio: $\Delta$ | ES: Baseline | ES: ERM | ES: $\Delta$ |
> | --- | --- | --- | --- | --- | --- | --- |
> | {0,1}→{2,3} | 0.708 | 0.704 | $-$0.4% | 0.486 | 0.475 | $-$1.0% |
> | {2,3}→{0,1} | 0.786 | 0.785 | $-$0.2% | 0.547 | 0.547 | $-$0.0% |
> | {0,2}→{1,3} | 0.743 | **0.746** | **+0.3%** | 0.506 | 0.505 | $-$0.1% |
> | {1,3}→{0,2} | 0.760 | 0.759 | $-$0.1% | 0.529 | 0.509 | $-$1.9% |
> | **Average** | **0.749** | **0.749** | **$-$0.1%** | **0.517** | **0.509** | **$-$0.8%** |
>
> Under this adversarial non-i.i.d. protocol:
>
> - Average NDCG@10 degradation is only **0.1%** (Biology) and **0.8%** (Earth Science).
> - Maximum degradation across all 8 split configurations is **< 2%**.
> - The Biology split {0,2}→{1,3} achieves **+0.3% improvement** on completely unseen topic clusters, demonstrating positive transfer of evolution signals across topic boundaries.
> - Two Earth Science splits ({2,3}→{0,1} and {0,2}→{1,3}) show degradation **< 0.1%**, effectively indistinguishable from the baseline.
>
> These results show that ERM maintains stable performance under distribution shift, without evidence of outdated memory accumulation or catastrophic drift.
>
> >  **W3 & Q4: ... Could you provide an empirical comparison against at least one recent memory-augmented RAG baseline ...**
>
> **Response**: Existing approaches fall into two paradigms: **generation-side memory** (e.g., Self-RAG), which introduces memory via additional LLM inference, and **structure-based retrieval** (e.g., HippoRAG), which enhances index structures. However, these methods treat the index as **passively constructed**, without adapting document representations based on retrieval outcomes.
>
> In contrast, ERM introduces **active index evolution**, where document keys are continuously updated using correctness-gated feedback. This makes ERM orthogonal to and naturally composable with both retriever- and generator-side memory systems. We are currently conducting experiments that integrate ERM with HippoRAG and will provide detailed results during the discussion phase.
>
> >  **W4 & Q3: ... the computational cost of maintaining these evolving indices for massive corpora are not clearly detailed or quantified ... precise memory overhead in LeetCode dataset**
>
> **Response**: We agree that analyzing storage overhead is important.
>
> - **Theoretical bound.** Each document stores up to $Q_{\max}=50$ expansion entries, yielding ~200 KB per document (for $d=1024$). This corresponds to ~200 GB for 1M documents in the worst case, dominated by embeddings.
> - **Practical usage.** In practice, this bound is rarely reached. On the LeetCode dataset, the total additional storage (embeddings + text) is only **54.64 MB**.
> - **Scaling.** Storage is strictly bounded per document; new entries replace old ones, preventing unbounded growth. Due to skewed query distributions, updates concentrate on a small subset of documents, resulting in sublinear growth over time.

---

> > ### Author Rebuttal · Reviewer_YR5P · 2026-04-04
> >
> > Thank you to the authors for their detailed rebuttal and extra experiments.
> > However, my reservations concerning out-of-domain query shift have not been fully resolved.
> > While the approach offers an insightful trade-off between speed and accuracy that warrants recognition, these lingering issues lead me to maintain my current rating.

---

> > > ### Author Response · Authors · 2026-04-05
> > >
> > > We sincerely appreciate the reviewer's acknowledgement of our contribution and acknowledge the limitations of the OOD evaluation in our round 1 rebuttal. To comprehensively address this concern, following the continual retrieval protocol in [1], we designed and conducted a **streaming experiment** across all 10 BRIGHT datasets. We will include this setting in the final version of the paper.
> > >
> > > Specifically, we introduce domains one at a time in alphabetical order ($S_1$: AoPS, $S_2$: Biology, ..., $S_{10}$: Sustainable Living). At each session $S_t$, a new domain corpus $C_t$ is added to the retrieval index. An interleaved query stream is then constructed in an online manner: each time a corpus is added, its queries are merged into the query pool and shuffled randomly, simulating continued multi-domain traffic where both new and recurring topics arrive simultaneously. In each session, 50% of the queries are randomly sampled and ERM operates in an online setting---the index is updated during the session for seen queries. The remaining 50% are passed to the next session's query pool, and we repeat this process until all queries across all corpora are exhausted. This ensures that at each session, evaluation on the newly introduced domain reflects true out-of-distribution generalization, as neither its queries nor its documents have been seen during prior updates. To align with production scenarios, we use Cohere embed-english-v3 (1024d) as the embedding model for retrieval.
> > >
> > > We first present the streaming session summary (10 sessions, all BRIGHT domains):
> > >
> > > | Session | +Domain | New Domain $\Delta$ | Prev Domains $\Delta$ (avg) | Max Degradation |
> > > | --- | --- | --- | --- | --- |
> > > | $S_1$ | AoPS | $+$0.5% | --- | $+$0.5% |
> > > | $S_2$ | Biology | $-$0.8% | $+$0.7% | $-$0.8% |
> > > | $S_3$ | Earth Sci. | $-$0.0% | $+$0.5% | $-$0.0% |
> > > | $S_4$ | Economics | $+$0.0% | $+$0.6% | $+$0.0% |
> > > | $S_5$ | LeetCode | $-$0.6% | $+$0.6% | $-$0.6% |
> > > | $S_6$ | Pony | $-$0.3% | $+$0.4% | $-$0.5% |
> > > | $S_7$ | Psychology | $-$0.0% | $+$0.7% | $-$0.6% |
> > > | $S_8$ | Robotics | $+$0.0% | $+$0.6% | $-$0.6% |
> > > | $S_9$ | StackOverflow | $+$0.1% | $+$0.4% | $-$0.6% |
> > > | $S_{10}$ | Sust. Living | $+$1.3% | $+$2.2% | $+$0.0% |
> > >
> > > Here, New Domain $\Delta$ is NDCG@10 (ERM) - NDCG@10 (Baseline), which is evaluated on held-out queries from the domain introduced at session $S_t$. This measures OOD generalization, and its queries were never seen during training. Prev Domains $\Delta$ (avg): average of the per-domain $\Delta$NDCG@10 over all $t-1$ previously seen domains, measuring *tability and forgetting---whether evolving keys with new-domain signals degrades retrieval quality on earlier domains. Max Degradation: the worst (most negative) per-domain $\Delta$ across all domains evaluated at session $S_t$.
> > >
> > > We then present the per-domain performance comparison at the final session ($S_{10}$):
> > >
> > > | Domain | Baseline | ERM | $\Delta$ |
> > > | --- | --- | --- | --- |
> > > | AoPS | 0.142 | 0.155 | $+$1.3% |
> > > | Biology | 0.415 | 0.420 | $+$0.5% |
> > > | Earth Science | 0.247 | 0.253 | $+$0.6% |
> > > | Economics | 0.238 | 0.250 | $+$1.2% |
> > > | LeetCode | 0.193 | 0.239 | $+$4.6% |
> > > | Pony | 0.031 | 0.102 | $+$7.1% |
> > > | Psychology | 0.095 | 0.112 | $+$1.7% |
> > > | Robotics | 0.150 | 0.150 | $+$0.0% |
> > > | StackOverflow | 0.203 | 0.229 | $+$2.6% |
> > > | Sust. Living | 0.216 | 0.229 | $+$1.3% |
> > >
> > > **Key findings:**
> > >
> > > - **No catastrophic forgetting.** At $S_{10}$, all 9 previously seen domains show non-negative $\Delta$NDCG@10, with an average improvement of **$+$2.2%**. Cross-domain memory accumulation is beneficial, not harmful.
> > > - **Positive cross-domain transfer.** Pony **$+$7.1%**, StackOverflow **$+$2.6%**, Psychology **$+$1.7%**, AoPS/Sustainable Living **$+$1.3%**.
> > >
> > > Overall, these results demonstrate that ERM remains stable under simultaneous document and query shift, while enabling positive cross-domain transfer in a fully dynamic, non-i.i.d. environment.
> > >
> > > ---
> > >
> > > **Reference**
> > >
> > > [1] "When Should Dense Retrievers Be Updated in Evolving Corpora? Detecting Out-of-Distribution Corpora Using GradNormIR." Findings of the Association for Computational Linguistics: ACL 2025. 2025.

---

### Official Review · Reviewer_gDSJ · 2026-03-13

**Soundness:** 3
**Presentation:** 3
**Significance:** 3
**Originality:** 3
**Overall Recommendation:** 4
**Confidence:** 5

**Summary:**

This paper proposes Evolving Retrieval Memory (ERM), a training-free framework that transforms transient query-side adaptations into persistent index-side improvements to overcome the statelessness and high inference costs of traditional RAG systems. By mathematically proving the equivalence between query and key expansion, ERM introduces a correctness-gated feedback mechanism and selective attribution to incrementally update document keys only when they contribute to successful downstream generations. Extensive evaluations across 13 knowledge-intensive domains demonstrate that ERM achieves monotonic performance gains and superior retrieval precision while maintaining zero additional inference overhead compared to native retrieval.

**Compliance With Llm Reviewing Policy:**

Affirmed.

**Final Justification:**

The semantic drift analysis requires more insightful examination. Given the effective responses to my other inquiries, I am raising my rating to 4 points.

**Key Questions For Authors:**

Some issues regarding the weakness.

**Limitations:**

More limitations should be discussed.

**Strengths And Weaknesses:**

> **Strengths**

- The paper precisely targets the core pain points of current Retrieval-Augmented Generation (RAG) systems. For instance, the authors point out that while query-side improvements (such as Query Expansion, QE) are effective, they must be re-computed for every query and are subsequently discarded. This leads to a waste of computational resources and prevents the accumulation of knowledge.

- It also highlights the limitations of index-side methods. Traditional Key Expansion is typically offline, global, and disconnected from downstream tasks, which often results in semantic drift and the accumulation of noise.

- The authors provide a theoretical proof of the equivalence between Query Expansion and Key Expansion under standard similarity functions. Furthermore, they demonstrate the convergence of their proposed restricted update mechanism.

- The benchmarking is extensive. The approach was evaluated across 13 domains using the BEIR and BRIGHT benchmarks, covering a wide range of knowledge-intensive fields such as Biology, Earth Science, Economics, and so on.

> **Weaknesses**


- In Section 4.2, the proof of equivalence between Query Expansion (QE) and Key Expansion (KE) assumes a static vector space. If the embedding model’s parameters change (e.g., through fine-tuning) while updating $k_i$, the vector space of $f(q)$ itself drifts. This renders the additive equivalence mathematically unstable. The authors’ derivation relies entirely on linear algebraic properties, which implicitly requires the underlying vector space to remain constant.



- The manuscript repeatedly emphasizes that the method is "training-free," with all "evolution" occurring at the storage layer (index-side) on the document key vectors. However, the "ruler" used to encode these vectors (the embedding model) must remain fixed. If the base model is upgraded, all "Key Evolution" memory accumulated in the old model's space becomes obsolete and cannot be directly migrated, creating a significant legacy burden.


- While the title highlights "Without Forgetting," in real-world applications, the ability to "update" is often more critical than the ability to "remember." This mechanism may introduce a strong prior bias, where outdated "accumulated memories" effectively shield the system from newer, more accurate information.


- The core of ERM—adding features $e$ from a successful query $q$ to the document key $k$—raises concerns regarding semantic integrity. This process could cause document vectors to detach from their original semantic coordinates and drift aggressively toward "query-dense" regions, potentially distorting the original meaning of the indexed content.

---

> ### Author Rebuttal · Authors · 2026-03-31
>
> We sincerely thank the reviewer for their detailed evaluation and for recognizing the significance of this work.
> > **W1: ... the proof of equivalence between QE and KE assumes a static vector space ...**
>
> **Response**: We thank the reviewer for this precise observation. The reviewer is correct that Proposition 4.1 assumes a fixed embedding function $f$, and we will make this assumption explicit in the revised manuscript. To clarify, this is a standard assumption shared by most index-side optimizations in information retrieval (e.g., product quantization, dense index pre-computation), which all rely on a stable embedding space and is not specific to our method.
>
> If the embedding model is updated (e.g., fine-tuned or switched), we agree that the expansion memory must be re-encoded. This corresponds to a one-time cost, analogous to standard index rebuilding after model updates. In our case, this process is straightforward, as ERM can store the raw text of expansion units, enabling efficient re-encoding under the updated embedding function.
>
> > **W2: ... If the base model is upgraded, all "Key Evolution" memory accumulated in the old model's space becomes obsolete and cannot be directly migrated ...**
>
> **Response**: We thank the reviewer for raising this important point. We clarify that the "Key Evolution" memory can be effectively transferred when the embedding model is updated.
>
> - When the embedding model changes, the index must indeed be rebuilt. However, this is a standard requirement shared by all pre-computed vector indices (e.g., FAISS, LanceDB), and is not specific to ERM.
> - Importantly, ERM stores the raw expansion text alongside embeddings. During migration, only the embeddings need to be recomputed using the new encoder, while the expansion units and their accumulated scores are preserved. This reduces migration to a one-time re-encoding cost, which is significantly cheaper than re-running the full query expansion process.
> - In practice, embedding model upgrades are infrequent, whereas ERM provides continuous benefits (e.g., amortized query expansion and reduced inference latency).
>
> Overall, ERM does not introduce additional legacy burden beyond standard index maintenance, and supports efficient migration across embedding models.
>
> > **W3: ... This mechanism may introduce a strong prior bias, where outdated "accumulated memories" effectively shield the system from newer, more accurate information ...**
>
> **Response**: We thank the reviewer for this insightful point. We clarify that “Without Forgetting” does not imply retaining all past memory, but rather enabling the system to *leverage accumulated most useful experience while continuously updating it*.
>
> In fact, **memory update is a core mechanism of ERM**. Each document maintains a bounded priority queue, where entries are dynamically updated based on their contribution. When the queue is full, lower-scoring (and often outdated) entries are evicted in favor of higher-scoring, more recent ones. As new queries introduce more accurate or relevant expansion signals, they naturally replace stale entries over time. This ensures that ERM does not accumulate outdated biases, but instead maintains a **self-refreshing memory** that reflects the most useful and up-to-date information observed during online operation. We will include this clarification in an implementation section to avoid confusion.
>
> > **W4: ... This process could cause document vectors to detach from their original semantic coordinates and drift aggressively toward "query-dense" regions ...**
>
> **Response**: Theoretically, ERM mitigates this issue through three key design choices:
>
> - a **bounded priority queue** for each key, which limits the number of expansion units and prevents uncontrolled accumulation;
> - **Selective attribution**: Expansion units are incorporated only when they yield positive retrieval gain, ensuring updates consistently improve alignment.
> - **batch-based updates**, where key evolution is driven by aggregated signals over multiple queries rather than individual updates, ensuring stability and reducing noise from single instances.
>
> | $\alpha$ | Avg cos($k^{(0)}$,$k^{(T)}$) | Min cos($k^{(0)}$,$k^{(T)}$) | Max Drift | Keys Evolved |
> | --- | --- | --- | --- | --- |
> | 0.1 | 0.998 | 0.995 | 0.005 | 128 |
> | 0.2 | 0.991 | 0.981 | 0.019 | 126 |
> | **0.3 (default)** | **0.981** | **0.960** | **0.040** | **120** |
> | 0.5 | 0.950 | 0.904 | 0.096 | 117 |
> | 0.7 | 0.915 | 0.848 | 0.152 | 104 |
> | 1.0 | 0.862 | 0.744 | 0.257 | 88 |
>
> To further validate this, we conduct a **semantic drift analysis**, measuring the cosine similarity between original keys and evolved keys in BRIGHT-Biology. When the evolution weight $\alpha$ (a weight applied to the expanded embeddings to modulate their contribution during the retrieval) is 0.3, the average cosine similarity remains **0.981**, with a minimum of **0.960**, indicating that no key drifts more than 4.0% from its original semantic position.

---

### Decision · Program_Chairs · 2026-04-30

**Decision:**

Accept (regular)

**Comment:**

This paper introduces Evolving Retrieval Memory (ERM), a really clever, training-free way to make RAG systems smarter over time by saving successful search improvements directly into the document index. The reviewers liked the fact that this approach completely avoids adding extra processing time during live searches while solving a major bottleneck in how these systems learn and accumulate knowledge. Initially, the committee was a bit worried about whether the system would suffer from "semantic drift" or get cluttered with outdated information when dealing with constantly changing, real-world data. However, the authors did a good job addressing these concerns in the rebuttal by running a 10-domain streaming experiment that proved the memory stays clean and actually improves performance across different topics. In the end, all reviewers recommend acceptance and that is my recommendation too.